# $\mathcal{RTV}$-$Bench$: Benchmarking MLLM Continuous Perception, Understanding and Reasoning through $\mathcal{R}$eal-$\mathcal{T}$ime $\mathcal{V}$ideo

**Shuhang Xun**[1]*, **Sicheng Tao**[2]*, **Jungang Li**[2,3]*†, **Yibo Shi**[4], **Zhixin Lin**[5],
**Zhanhui Zhu**[1], **Yibo Yan**[2,3], **Hanqian Li**[2], **Linghao Zhang**[5],
**Shikang Wang**[6], **Yixin Liu**[1], **Hanbo Zhang**[7], **Ying Ma**[1]‡, **Xuming Hu**[2,3]

[1] HIT    [2] HKUST (GZ)    [3] HKUST    [4] XJTU    [5] SDU    [6] CityU    [7] HUST

**Project:** https://ljungang.github.io/RTV-Bench

## Abstract

Multimodal Large Language Models (MLLMs) have made rapid progress in perception, understanding, and reasoning, yet existing benchmarks fall short in evaluating these abilities under continuous and dynamic real-world video streams. Such settings require models to maintain coherent understanding and reasoning as visual scenes evolve over time. We introduce $\mathcal{RTV}$-$Bench$, **a fine-grained benchmark for real-time video analysis with MLLMs**. It is built upon three key principles: multi-timestamp question answering, hierarchical question structures spanning perception and reasoning, and multi-dimensional evaluation of continuous perception, understanding, and reasoning. $\mathcal{RTV}$-$Bench$ comprises 552 diverse videos and 4,608 carefully curated QA pairs covering a wide range of dynamic scenarios. We evaluate a broad range of state-of-the-art MLLMs, including proprietary, open-source offline, and open-source real-time models. Our results show that real-time models generally outperform offline counterparts but still lag behind leading proprietary systems. While scaling model capacity generally yields performance gains, simply increasing the density of sampled input frames does not consistently translate into improved results. These observations suggest inherent limitations in current architectures when handling long-horizon video streams, underscoring the need for models explicitly designed for streaming video processing and analysis.

## 1 Introduction

The ability to comprehend and respond to complex real-world scenarios in real time remains a fundamental challenge in the pursuit of general artificial intelligence [6, 25, 10]. Motivated by the remarkable success of large language models (LLMs) across a broad spectrum of tasks [26, 11, 2], multimodal large language models (MLLMs) have recently emerged as a promising paradigm for visual scene understanding and reasoning [49, 42, 5]. In particular, the research trajectory of Video-LLMs [22, 28, 4, 36, 34] has evolved from early studies focused on short, vision-centric video clips [16, 14, 23, 20] toward more comprehensive modeling of long-form video content. An increasing body of work integrates omni-modal signals—including video, audio, and subtitles [3, 7, 18, 33, 19, 27]—to support richer contextual understanding and more robust long-horizon reasoning.

---

*Equal contribution. Emails: Shuhang Xun (24s103400@stu.hit.edu.cn)

†Project Leader. Emails: ljungang.02@gmail.com.

‡Corresponding author. Email: y.ma@hit.edu.cn.

39th Conference on Neural Information Processing Systems (NeurIPS 2025) Track on Datasets and Benchmarks.

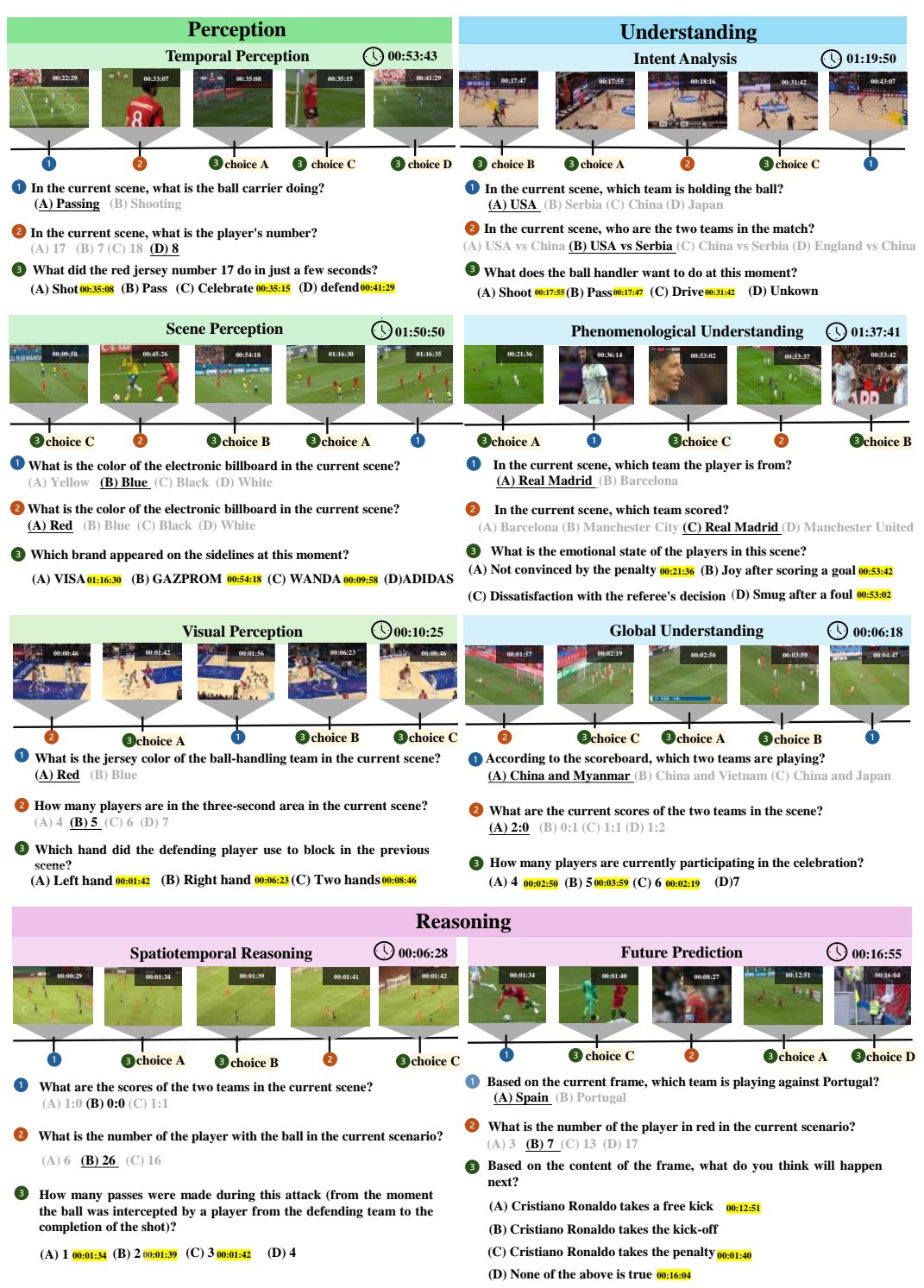

Figure 1: **Representative examples illustrating the diverse task types evaluated in** $\mathcal{RTV}$-*Bench*. ① and ② denote fundamental questions within a question group, with their corresponding answers underlined. ③ indicates a dynamically answered question, where the correct response is determined by the query time. As the visual content evolves, the correct answer may change over time; we therefore annotate the appropriate answers corresponding to different query timestamps.

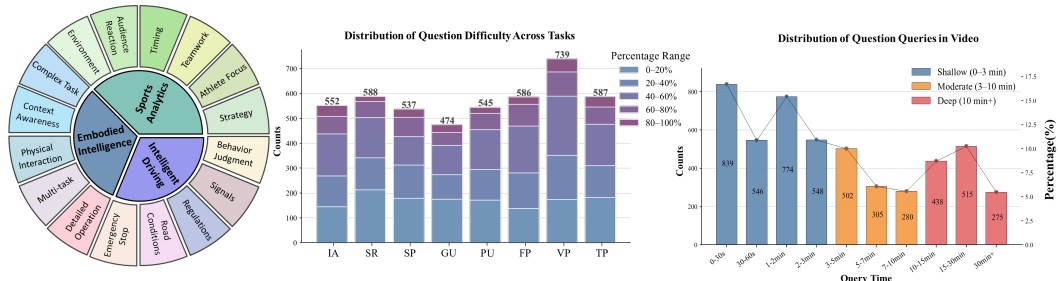

Figure 2: **Video categories and distributions of question difficulty and query characteristics.** (Left) RTV-Bench covers three key domains and 16 video subcategories. (Center) Distribution of question difficulty levels across eight representative task types, quantified by percentage-based performance ranges. (Right) Distribution of question queries with respect to video length, categorized into Shallow, Moderate, and Deep levels. Bar heights indicate query counts, while the overlaid line chart shows the proportion of queries within each duration bucket.

Recently, VStream [46] was the first to attempt to test this capability, with a focus primarily on extending the duration of videos. In addition, both StreamingBench [40] and OVOBench [21] have made varying degrees of improvements in the types and standards of assessment. However, their evaluation of real-time responsiveness is often inadequate, overlooking the capacity to capture transitions and fleeting details from visual input that arrive sequentially, not instantaneously. This limitation highlights the need for a more focused assessment of continuous analysis abilities.

Based on the above considerations, we introduce $\mathcal{RTV\text{-}Bench}$, featuring three core innovations designed to benchmark the **continuous analysis capabilities**—specifically perception, understanding, and reasoning—of MLLMs within real-time video contexts. First, the **Multi-Timestamp Q&A** mechanism challenges real-time tracking and state update by posing queries whose answers evolve within a video. Crucially, unlike benchmarks like OVO-Bench [21] that typically introduce different questions at different timestamps, RTV-Bench revisits the same conceptual query, where only the correct answer shifts as the scene unfolds. This approach more rigorously tests the model capacities for continuous analysis in real-time scenarios, surpassing single-query-single-answer and static evaluations. Second, the **Hierarchical Question Structure** enforces reliable, sequential reasoning by employing a basic-to-advanced design where higher-order questions logically depend on grasping foundational perceptions and understanding, thus mitigating cognitive shortcuts. Finally, **Multidimensional Evaluation** moves beyond aggregate scores to provide fine-grained diagnostic insights, assessing model performance across eight dimensions that are critical for continuous analysis in dynamic scenarios. This evaluation offers a more informative view of model capabilities and limitations in real-time video understanding; detailed examples are presented in Figure 1.

Through the novel design of $\mathcal{RTV\text{-}Bench}$ and the comprehensive evaluations presented in this work, we provide systematic insights into the current state of MLLMs for continuous video analysis. Results across all evaluated models reveal substantial bottlenecks in real-time video understanding: ❶ most models achieve accuracies below 50%; ❷ overall performance shows a clear positive correlation with model scale, whereas increasing the number of input frames yields only marginal and non-monotonic gains; and ❸ models explicitly designed for streaming video processing consistently outperform traditional offline video models. Notably, even the lowest-performing real-time model evaluated, VITA-1.5 [9], surpasses a representative offline counterpart, VideoLLaMA2 [4]. Building on these findings, we further discuss promising research directions for online video analysis with MLLMs.

## 2   Real-Time Video Understanding for MLLMs: RTV-Bench

In this section, we discuss a common challenge in both long video comprehension and real-time video analysis—the continuous analysis capability of MLLMs. To tackle this, we have meticulously curated a benchmark to assess model performance in real-time and continuous scenarios, and developed new metrics to characterize the performance of multimodal large models in this bench.

## 2.1 Challenge for Real-Time Video Analysis

Existing benchmarks for long video analysis [8, 39] attempt to gauge the true capacity of MLLMs by extending video duration, enhancing the difficulty of QA tasks, and introducing new sub-tasks. Other works [24] focus on real-time video scenarios, evaluating model responsiveness to queries in real-time contexts. However, current models face the risk of memory loss and attention shift, both in long and real-time video contexts. We categorize these challenges as failures in the continuous analysis capability of MLLMs. In video scenarios, MLLMs are considered to possess strong continuous analysis ability only if they can effectively recall prior visual information to accurately respond to user queries (similar to needle-in-a-haystack tasks in natural language settings) and maintain robust perception of streams of new visual data and queries (akin to multimodal multi-turn dialogues).

## 2.2 Benchmark Overview

The RTV-Bench is designed to assess a model's ability to perceive, understand, and reason in long video and real-time streaming contexts. Models should be adept at recognizing correct temporal query patterns. For example, during live sports queries (*e.g.*, goalkeeper actions), models must prioritize current contexts over historical data. However, it should also retrieve relevant information from memory, such as identifying the goalkeeper from earlier footage. This necessitates that models not only possess exceptionally high fundamental scene comprehension capabilities, but also demonstrate effective understanding of all stages within the temporal flow—namely, comprehensive spatiotemporal comprehension encompassing the past, present, and future. To this end, we propose a series of sub-tasks for video continuous analysis, addressing both spatiotemporal elements within the video and the intrinsic capabilities of the model. The intrinsic model capabilities are divided into eight categories: Temporal Perception (TP), Scene Perception (SP), Visual Perception (VP), Future Prediction (FP), Phenomenological Understanding (PU), Intent Analysis (IA), Global Understanding (GU), and Spatiotemporal Reasoning (SR). An overview of these task categories is provided in Figure 1.

## 2.3 Benchmark Construction

**Terminology** The RTV-Bench dataset comprises 552 videos sourced from the internet. A key characteristic is its question structure designed to directly probe temporal dynamics. For each video, questions are organized into sets. Each question set contains about three multiple-choice questions.

Crucially, the core design principle centers on time-varying correct answers through Multi-Timestamp QA. Specifically, within a set of questions, the same underlying conceptual query (*e.g.*, "What is person A holding?" or "Where is the car heading?") is evaluated at multiple points in time throughout the video. As a result, the correct answer may differ depending on the specific timestamp or temporal interval referenced—or implicitly required—by the question context or its earliest inferable timestamp (Figure 1). Rather than merely locating relevant information, models are therefore required to actively track temporal changes and continuously update their understanding as the scene evolves. This design directly targets the evaluation of continuous temporal understanding and a model's sensitivity to dynamic state transitions; representative examples are shown in Figure 4.

The question set features three questions of escalating difficulty. The first two are simpler, while the final question is significantly more complex, demanding integration of broader context or clues. The information and reasoning needed for the final question generally encompass those required for the first two. Successfully answering the complex third question strongly suggests the capability to also answer the simpler ones. This structure evaluates the model's ability to handle increasing complexity, synthesize comprehensive context, and perform robust analysis.

In addition, we emphasize diversity through the richness of specific sub-scenes and the distribution of video lengths. The RTV-Bench primarily encompasses intelligent driving, sports events, and egocentric videos—categories rich in dynamic information and real-time contextual relevance. Each category includes a variety of sub-categories, as illustrated in Figure 2.

**Data Statistic** RTV-Bench comprises 552 videos with a total duration of 167.2 hours (average 18.2 minutes per video) and contains 4,631 QA pairs, for detailed information, refer to Figure 2 and Table 1. The benchmark features diverse question scenarios and evenly distributed video durations, thereby establishing a comprehensive framework for video continuous understanding tasks with rich sub-scenarios and structured problem dimensions.

Table 1: **Comparison of video QA benchmarks.** MT denotes multi-timestamp questions. Labels: A (automatic), M (manual), A+M (both).

| Benchmark | MT | #QA (k) | Avg. Duration (min) | Total Duration (h) | Labels |
|---|---|---|---|---|---|
| Video-MME [8] | ✗ | 2.7 | 17.00 | 254.0 | M |
| MSRVTT [41] | ✗ | 73.0 | 0.25 | 12.5 | A |
| MSVD-QA [1] | ✗ | 13.0 | 0.17 | 1.4 | A |
| MovieChat-1k [32] | ✗ | 13.0 | 9.40 | 156.0 | M |
| MVBench [17] | ✗ | 4.0 | 0.25 | 34.5 | A+M |
| ActivityNet-QA [44] | ✗ | 58.0 | 1.87 | 25.0 | M |
| Vstream-Q [46] | ✗ | 3.5 | 40.00 | 21.0 | – |
| StreamBench [38] | ✗ | 1.8 | 4.50 | 25.0 | M |
| OVO-Bench [21] | ✗ | 2.8 | 6.00 | 66.0 | A+M |
| OVBench [12] | ✓ | 7.0 | – | – | A |
| **RTV-Bench (Ours)** | ✓ | 4.6 | 18.00 | 167.2 | M |

**Video Collection and Filtering**    Our video data sources include EgoSchema [29] and publicly available online videos. Unlike most existing benchmarks, we incorporated manual review during the collection phase. Three data collectors sourced videos from various domains and manually excluded highly similar videos, focusing on long videos with high dynamics and real-time needs. These sources ensure targeted and scientific evaluation of video models' real-time capabilities.

**Manual Annotation**    Rigorous manual annotation by qualified experts underpins the reliability of RTV-Bench for evaluating continuous video understanding. We leverage an LLM (DeepSeek [26]) only to produce initial question templates, and human annotators then refine every question to better reflect dynamic scenes and the demands of temporal reasoning. In particular, annotators intentionally craft questions whose correct answers evolve over time and systematically determine the earliest valid timestamp associated with each answer option in the MTQA setting. This human-centric, multi-annotator protocol strengthens annotation robustness and enables RTV-Bench to explicitly evaluate models' sensitivity to temporal dynamics in video.

**Quality Control**    To ensure the benchmark's quality, each video and its corresponding Q&A pairs underwent multiple rounds of review. We manually filtered videos based on length distribution and the presence of sub-scenes examining real-time event changes, resulting in high-quality videos focused on real-time analysis tasks. We conducted manual video-question alignment and precise timestamp checks, utilizing GPT-4 and human review to verify annotation format and sensitive information.

## 3   Experiments

### 3.1   Experiment Setup

Our experiments were conducted on two NVIDIA A800 GPUs to comprehensively evaluate the performance of mainstream multimodal large language models (MLLMs) on our benchmark. We consider a diverse set of representative models, including VideoLLaMA2 [4], VideoLLaMA3 [45], GPT-4o [13], InternLM-XComposer2.5-OmniLive (IXC2.5-OL) [47], VITA1.5 [9], VideoChat-Online (4B) [12], LLaVA-Video [22], LLaVA-OneVision [15], and Qwen2.5-VL [43].

To ensure a fair comparison under comparable computational budgets, most models are evaluated using configurations around the 7B scale when available, while smaller models (e.g., VideoChat-Online at 4B) are evaluated at their native parameter size. All models are tested under a unified uniform frame sampling protocol. Specifically, for models that support variable frame inputs (e.g., Qwen2.5-VL), we evaluate multiple sampling settings with 8, 16, 32, and 64 uniformly sampled frames. For each model, we report in the main tables the best-performing configuration across different frame counts, following the same evaluation protocol for all baselines. Other models are evaluated analogously using their supported frame sampling ranges, and their strongest results are reported for comparison.

**Real-Time Video Model vs. Offline Video Model**    We compare two model categories: traditional offline models ($M_{offline}$) and novel real-time online models ($M_{online}$). These categories differ significantly in architecture, training, and data requirements. Architecturally, $M_{offline}$ typically employ

sequential vision encoder-decoders, often limited by fixed context windows and incurring higher processing latency. In contrast, $M_{\text{online}}$ are designed for continuous, low-latency ingestion of the video stream $V$. They prioritize maintaining an internal state $S_t$ that summarizes the video information processed up to time $t$ (*i.e.*, $V[0, t]$), enabling real-time responsiveness. Models like IXC2.5-OL [47] implement this using techniques like modular parallelism and dedicated long-term memory. These architectural distinctions lead to different training paradigms and data needs. $M_{\text{offline}}$ usually rely on end-to-end fine-tuning using standard annotated video datasets. $M_{\text{online}}$, however, often require specialized training strategies (*e.g.*, VITA-1.5's staged fusion, IXC2.5-OL's targeted training for memory and interaction) and benefit most from specialized corpora designed for long-duration, interactive streaming scenarios. The ability of $M_{\text{online}}$ to maintain and update state $S_t$ is particularly relevant for RTV-Bench's Multi-Timestamp QA (MTQA) challenge, where the correct answer $A^*(Q, t_q)$ to a query $Q$ depends on the specific query time $t_q$. To evaluate both model types on this benchmark, we adapt the testing procedure. For $M_{\text{online}}$, queries $Q$ are presented at their timestamp $t_q$, and the models leverage their continuously updated state $S_{t_q}$ to generate the answer $A_{M_{\text{online}}} = M_{\text{online}}(Q, t_q | S_{t_q})$. Since $M_{\text{offline}}$ lack this inherent streaming capability, we simulate real-time interaction for them: when a query $Q_i$ is posed at time $t_{q,i}$, we extract and provide only the relevant video segment $V_i$ (corresponding to the query's context). The offline model's answer is thus based solely on this isolated segment: $A_{M_{\text{offline}}, i} = M_{\text{offline}}(Q_i, V_i)$.

To evaluate the performance of these models, we employed two metrics: **Accuracy** and **Score**.

**Accuracy.** The accuracy metric measures the proportion of correct answers provided by the model compared to the ground truth.

**Score.** The score metric evaluates the model's ability to correctly answer advanced-level questions (type q2), contingent upon its demonstrated mastery of prerequisite basic questions (types q0 and q1) within the same question group, thus emphasizing reliable advanced reasoning built upon a solid foundation. Calculation involves a prerequisite check for each group $i$: if all basic questions are correct ($B_i = 1$), the group contributes points equal to the number of correctly answered q2 questions ($N_{q2,i}^{\text{correct}}$); otherwise ($B_i = 0$), it contributes zero points. The final score is the ratio of total conditionally awarded points to the total number of q2 questions across all $N$ valid groups (those containing q2). Formula:

$$\text{Score} = \frac{\sum_{i=1}^{N} B_i \cdot N_{q2,i}^{\text{correct}}}{\sum_{i=1}^{N} N_{q2,i}^{\text{total}}}$$

where $N$ is the number of valid groups, $B_i$ is the prerequisite indicator (1 if basics are correct, 0 otherwise) for group $i$, $N_{q2,i}^{\text{correct}}$ is the count of correct q2 answers in group $i$, and $N_{q2,i}^{\text{total}}$ is the total count of q2 questions in group $i$. Advantages of this metric include: ensuring foundational accuracy by rewarding advanced correctness only when basics are mastered; reflecting model robustness by penalizing superficial success on complex tasks without fundamental understanding; and aligning with hierarchical learning principles where complex skills build upon simpler ones.

Table 2: Evaluation results on RTV-Bench. **Perception**, **Understanding**, and **Reasoning** denote different task categories. **FQA** refers to foundational video question answering without multi-timestamp supervision. **MTQA** refers to multi-timestamp question answering with time-varying correct answers. Scores are computed using group-aware Q2 evaluation.

| Model | #Size | Perception Acc (%) / Score | Understanding Acc (%) / Score | Reasoning Acc (%) / Score | FQA Acc (%) | MTQA Acc (%) | Overall Acc (%) / Score |
|---|---|---|---|---|---|---|---|
| *Open-Source Offline Video Models* | | | | | | | |
| Qwen2.5-VL [43] | 7B | 42.30 / 7.70 | 39.85 / 7.00 | 38.16 / 6.90 | 44.07 | 37.46 | 40.41 / 7.13 |
| VideoLLaMA2 [4] | 7B | 40.62 / 8.67 | 39.85 / 7.77 | 37.49 / 6.75 | 45.77 | 34.95 | 39.55 / 7.90 |
| VideoLLaMA3 [45] | 7B | 37.98 / 5.83 | 35.29 / 5.73 | 35.78 / 6.80 | 38.62 | 34.91 | 36.42 / 6.10 |
| LLaVA-OneVision [15] | 7B | 35.38 / 3.97 | 34.21 / 4.63 | 33.57 / 4.95 | 35.80 | 33.58 | 34.49 / 4.40 |
| LLaVA-Video [22] | 7B | 35.83 / 5.03 | 33.81 / 3.77 | 35.15 / 5.75 | 36.28 | 34.17 | 34.90 / 4.80 |
| *Open-Source Online Models* | | | | | | | |
| VITA-1.5 [9] | 7B | 45.66 / 12.80 | 44.12 / 11.83 | 43.37 / 10.15 | 55.06 | 36.32 | 44.51 / 11.80 |
| IXC2.5-OL [47] | 7B | 47.21 / 15.87 | 48.22 / 15.23 | 46.18 / 14.45 | **59.05** | 38.21 | 47.33 / 15.40 |
| VideoChat-Online [12] | 4B | 46.86 / 12.30 | 46.34 / 12.80 | 43.53 / 11.00 | 55.16 | 38.21 | 45.83 / 12.10 |
| *Closed-Source Business Models* | | | | | | | |
| GPT-4o [13] | – | **51.61** / **21.90** | **49.31** / **20.76** | **48.71** / **23.95** | 56.53 | **44.73** | **50.02** / **22.10** |
| Gemini 2.0 Flash [35] | – | 41.67 / 11.00 | 42.71 / 12.73 | 41.44 / 12.05 | 47.49 | 38.64 | 42.00 / 12.00 |

## 3.2 Experiment Results

**Online vs. Offline Models.** As shown in Table 2, online models optimized for real-time processing—particularly IXC2.5-OL—surpass offline counterparts in overall performance metrics. IXC2.5-OL achieves 47.33% Accuracy and 15.40 Score, significantly outperforming offline models like VideoLLaMA2 (39.55% Accuracy / 7.90 Score). Furthermore, when comparing online models, IXC2.5-OL demonstrates a clear advantage over VITA-1.5 with improvements of 2.82% in Accuracy and 3.6 points in Score. A notable performance gap emerges in temporal analysis tasks: IXC2.5-OL attains 38.21% Accuracy in Multi-Timestamp Question Answering (MTQA), substantially higher than the 33–35% range typical of offline models. This notable discrepancy suggests that current online models may be promising avenues toward continuous analysis capabilities.

Table 3: Detailed evaluation results on the category of **Perception**. Temporal Perception (TP), Visual Perception (VP) and Scene Perception (SP).

| Method | #Size | TP Acc (%) / Score | VP Acc (%) / Score | SP Acc (%) / Score | Overall Acc (%) / Score |
|---|---|---|---|---|---|
| *Open-Source Offline Video Models* | | | | | |
| Qwen2.5-VL [43] | 7B | 39.35 / 6.7 | 45.47 / 9.1 | 41.15 / 6.9 | 42.30 / 7.7 |
| VideoLLaMA2 [4] | 7B | 39.52 / 7.9 | 42.49 / 9.4 | 39.85 / 8.7 | 40.62 / 8.67 |
| VideoLLaMA3 [45] | 7B | 37.82 / 6.1 | 39.24 / 7.6 | 36.87 / 3.8 | 37.98 / 5.83 |
| LLaVA-OneVision [15] | 7B | 35.09 / 3.9 | 35.86 / 3.8 | 35.20 / 4.2 | 35.38 / 3.97 |
| LLaVA-Video [22] | 7B | 34.07 / 4.8 | 38.97 / 5.8 | 34.45 / 4.5 | 35.83 / 5.03 |
| *Open-Source Online Models* | | | | | |
| VITA-1.5 [9] | – | 46.51 / 12.1 | 47.09 / 13.2 | 43.39 / 13.1 | 45.66 / 12.8 |
| IXC2.5-OL [47] | 7B | 49.57 / 17.6 | 49.80 / 16.5 | 42.27 / 13.5 | 47.21 / 15.87 |
| VideoChat-Online [12] | 4B | 48.55 / 13.3 | 48.58 / 14.43 | 42.64 / 8.3 | 46.86 / 12.3 |
| *Closed-Source Business Models* | | | | | |
| GPT-4o [13] | – | 48.60 / 18.2 | 53.59 / 23.4 | 52.63 / 24.1 | 51.61 / 21.90 |
| Gemini 2.0 Flash [35] | – | 40.49 / 9.5 | 45.19 / 16.1 | 39.34 / 7.4 | 41.67 / 11.00 |

Table 4: Detailed evaluation results on the category of **Understanding**. Phenomenological Understanding (PU), Global Understanding (GU) and Intent Analysis (IA).

| Method | #Size | GU Acc (%) / Score | PU Acc (%) / Score | IA Acc (%) / Score | Overall Acc (%) / Score |
|---|---|---|---|---|---|
| *Open-Source Offline Video Models* | | | | | |
| Qwen2.5-VL [43] | 7B | 36.92 / 5.8 | 42.02 / 6.9 | 40.22 / 8.2 | 39.85 / 7.00 |
| VideoLLaMA2 [4] | 7B | 37.34 / 7.6 | 42.21 / 9.5 | 40.92 / 6.2 | 39.85 / 7.77 |
| VideoLLaMA3 [45] | 7B | 33.54 / 5.8 | 39.13 / 5.9 | 33.39 / 4.3 | 35.35 / 5.33 |
| LLaVA-OneVision [15] | 7B | 32.07 / 4.3 | 33.51 / 3.0 | 37.06 / 6.6 | 34.21 / 4.63 |
| LLaVA-Video [22] | 7B | 29.42 / 2.5 | 35.69 / 3.9 | 36.33 / 4.9 | 33.81 / 3.77 |
| *Open-Source Online Models* | | | | | |
| VITA-1.5 [9] | 7B | 40.30 / 7.2 | 46.01 / 15.1 | 46.06 / 13.2 | 44.12 / 11.83 |
| IXC2.5-OL [47] | 7B | 43.88 / 11.9 | 52.17 / 18.7 | 48.62 / 15.1 | 48.22 / 15.23 |
| VideoChat-Online [12] | 4B | 42.19 / 8.3 | 48.99 / 13.82 | 47.28/ 15.7 | 46.34 / 12.8 |
| *Closed-Source Business Models* | | | | | |
| GPT-4o [13] | – | 45.02 / 15.7 | 54.32 / 25.8 | 48.58 / 20.8 | 49.31 / 20.76 |
| Gemini 2.0 Flash [35] | – | 35.70 / 10.6 | 45.65 / 11.3 | 46.78 / 16.3 | 42.71 / 12.73 |

**Open-Source vs. Close-Source Models.** While a performance gap persists compared to leading closed-source models like GPT-4o, state-of-the-art online architectures demonstrate remarkable progress. The online model IXC2.5-OL achieves near-top-tier performance with 47.33% Overall Accuracy and 15.40 Score, substantially outperforming mid-range closed-source systems like Gemini 2.0 Flash. Notably, IXC2.5-OL closes the accuracy gap with GPT-4o to 4.4% in perception tasks (Tables 3) and 1.1% in video understanding, demonstrating competitive performance in multimodal

Table 5: Detailed evaluation results on the category of **Reasoning**. Future Prediction (FP) and Spatiotemporal Reasoning (SR).

| Method | #Size | FP Acc (%) / Score | SR Acc (%) / Score | Overall Acc (%) / Score |
|---|---|---|---|---|
| *Open-Source Offline Video Models* | | | | |
| Qwen2.5-VL [43] | 7B | 42.49 / 9.7 | 33.84 / 4.2 | 38.16 / 6.90 |
| VideoLLaMA2 [4] | 7B | 41.47 / 7.5 | 33.50 / 6.0 | 37.49 / 6.75 |
| VideoLLaMA3 [45] | 7B | 38.05 / 6.9 | 33.84 / 3.9 | 35.95 / 5.40 |
| LLaVA-OneVision [15] | 7B | 38.23 / 7.2 | 28.91 / 2.7 | 33.57 / 4.95 |
| LLaVA-Video [22] | 7B | 39.08 / 9.1 | 31.22 / 2.4 | 35.15 / 5.75 |
| *Open-Source Online Models* | | | | |
| VITA-1.5 [9] | 7B | 47.95 / 12.2 | 38.78 / 8.1 | 43.37 / 10.15 |
| IXC2.5-OL [47] | 7B | 51.88 / 18.1 | 40.48 / 10.8 | 46.18 / 14.45 |
| VideoChat-Online [12] | 4B | 48.12 / 14.69 | 38.95 / 7.5 | 43.53 / 11.0 |
| *Closed-Source Business Models* | | | | |
| GPT-4o [13] | – | 54.67 / 27.1 | 42.75 / 20.8 | 48.71 / 23.95 |
| Gemini [35] 2.0 Flash | – | 44.42 / 13.6 | 38.46 / 10.5 | 41.44 / 12.05 |

domains. However, limitations emerge in complex reasoning where GPT-4o maintains decisive advantages, especially on complex tasks like **Understanding** and **Reasoning** (Tables 4 and 5). This pattern highlights that while modern online models have approached entry-level commercial systems and even challenged premium models in specific competencies, structural innovations remain critical to bridge gaps in advanced cognitive tasks like multi-step reasoning and temporal analysis.

**Impact of Model Scales.** We analyze the effect of model scale by evaluating Qwen2.5-VL from 3B to 72B parameters under different frame sampling budgets (8–64 frames), as shown in Figure 3(c). Overall accuracy exhibits a clear and largely monotonic improvement with increasing model size. Specifically, the 72B model consistently achieves the highest performance across all frame settings, reaching up to 40.78% with 64 frames, while smaller models (3B–32B) remain below 40%.

Notably, scaling benefits are consistent but moderate, with absolute gains from 3B to 72B on the order of ∼2–3 points, suggesting diminishing returns at larger scales. In addition, model scale interacts with temporal resolution: larger models benefit more reliably from increased frame counts, whereas smaller models show non-uniform or even fluctuating trends when additional frames are introduced. These observations indicate that while parameter scaling remains beneficial for real-time video understanding, its effectiveness is increasingly constrained by architectural and temporal modeling

capacities, highlighting the importance of improving temporal representation efficiency beyond naive model enlargement.

**Impact of Frame Numbers.** Figure 3(c,d) jointly examine the effect of frame sampling density across model scales and architectures. From Figure 3(c), increasing the number of frames from 8 to 64 does not yield consistent accuracy gains across model sizes. While larger models (*e.g.*, 72B) show modest improvements with more frames, smaller and medium-scale models exhibit non-monotonic or saturated trends, indicating limited benefit from denser temporal sampling. In some cases (*e.g.*, 3B and 32B), additional frames lead to marginal gains or even slight regressions.

This phenomenon becomes more pronounced in Figure 3(d), which aggregates performance across models. Average accuracy remains largely stable as frame count increases, while the total score—reflecting global understanding—often declines. Notably, IXC2.5-OL suffers a clear performance drop with more frames, suggesting that excessive temporal inputs may overwhelm the model's effective processing capacity. Together, these results indicate that simply increasing frame numbers is insufficient for improving real-time video understanding and may instead introduce redundancy or attention dilution. This highlights the need for temporally selective, adaptive frame utilization strategies rather than uniform increases in sampling density.

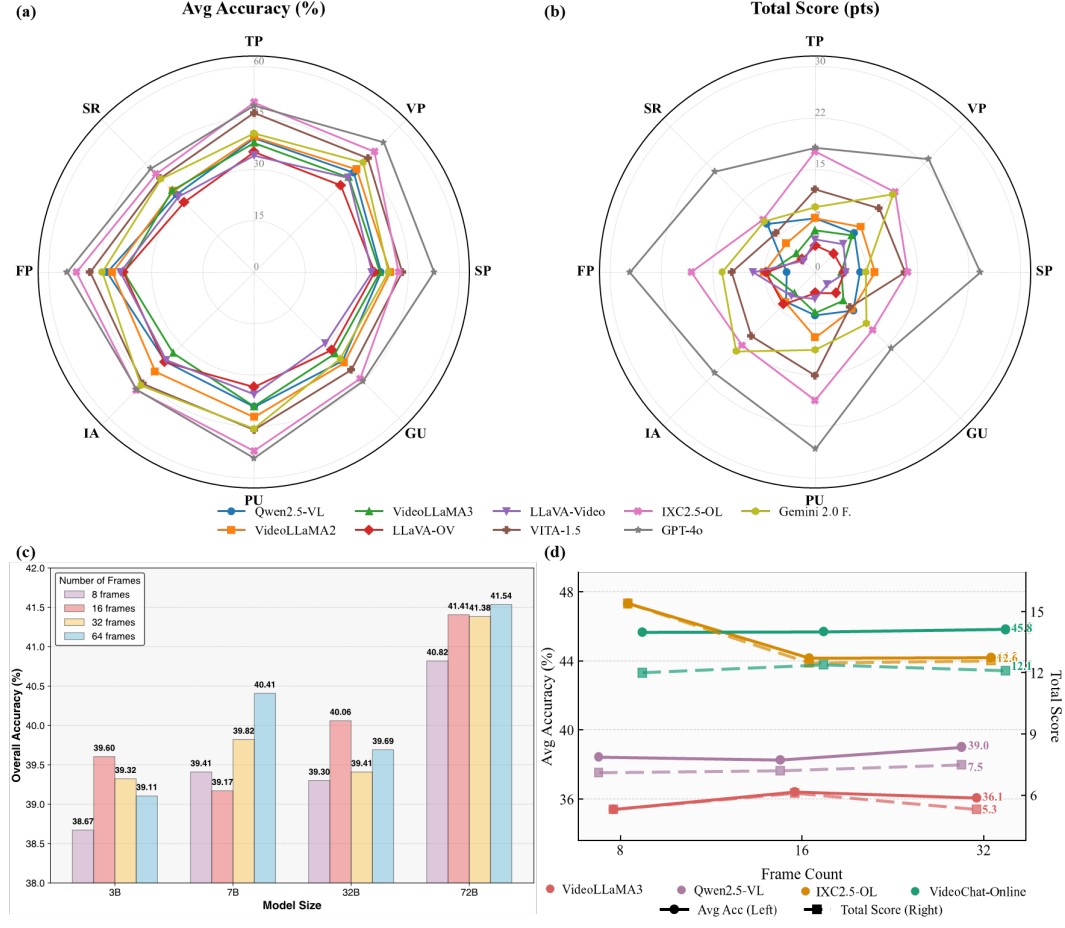

Figure 3: **Performance visualization and analysis on RTV-Bench**: (a) Visualization of overall **Accuracy** results; (b) Visualization of overall **Score** results; (c) Performance impact of varying input frame counts; (d) Performance comparison across different Qwen2.5-VL model scales.

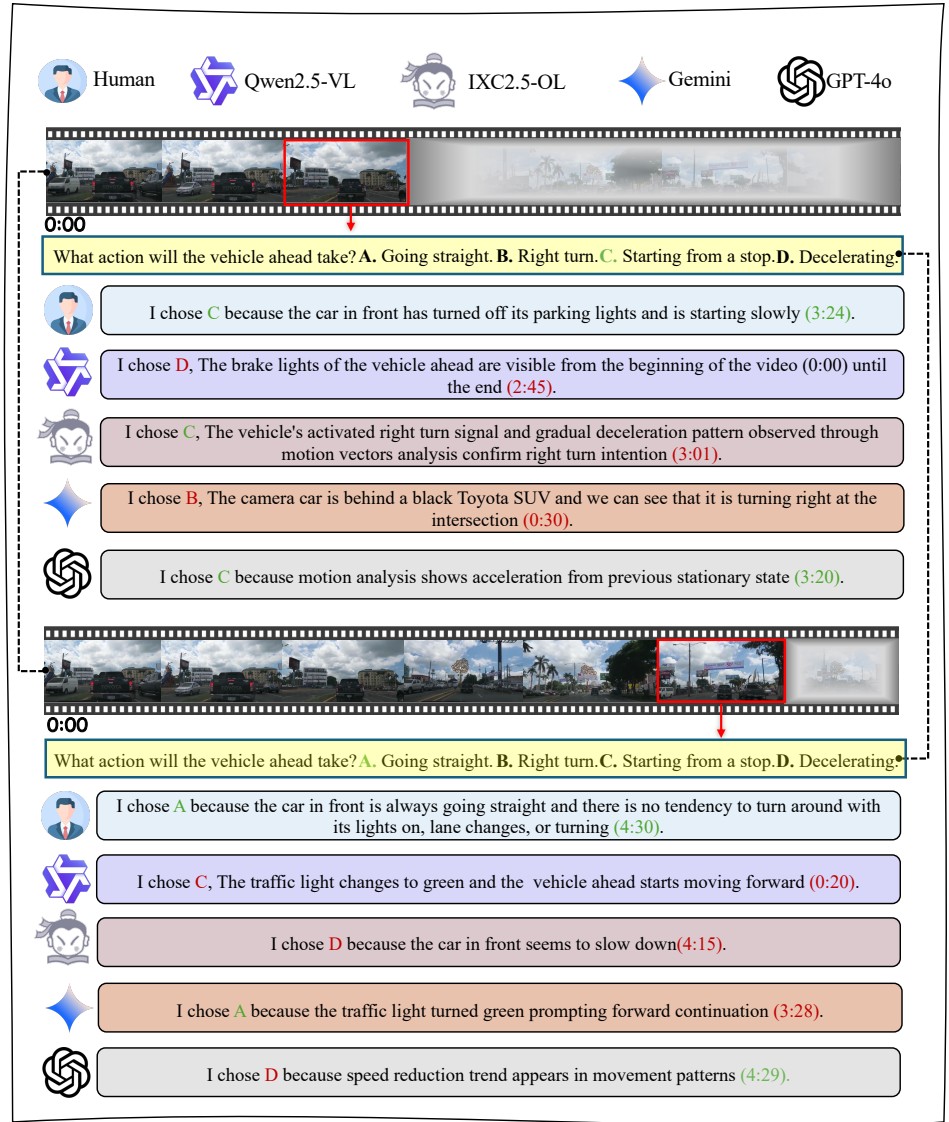

Figure 4: **Comparison of response from different models for the same question on the same video.** Green indicates correct answers or timestamps; red indicates incorrect answers or timestamps. This case demonstrates that even current high-performance models struggle to provide high-quality responses (where both the answer and the corresponding timestamp are accurate).

# 4 Related Work

## 4.1 Advanced MLLMs towards Real-time Video Analysis

Video-oriented large-scale models represent a highly promising domain with significant real-world potential, giving rise to numerous valuable applications [28, 8]. Video-LLMs have rapidly advanced, moving from analyzing short videos [16, 22] to longer ones [31, 36, 48] using various techniques. However, most research targets offline video analysis. Real-world applications necessitate real-time, continuous perception and reasoning on unfolding video streams. While models like InternLM-XComposer2.5-OmniLive [47], VITA-1.5 [9], and Dispider [30] are exploring real-time capabilities, rigorously evaluating their ability to continuously track and understand dynamic events remains a critical challenge.

## 4.2 Existing Video Benchmarks

Existing benchmarks primarily focus on offline evaluation using offline videos [8, 37, 39]. These are less suited for assessing how MLLMs track dynamically changing states, as they often use static question-answer pairs. Newer benchmarks tackle real-time and streaming aspects [24, 21], evaluating responsiveness and contextual understanding in online settings. However, RTV-Bench specifically addresses the gap in evaluating continuous perception, understanding, and reasoning. Its key distinction is the use of dynamic question answering, where the correct answer evolves with the video stream, directly probing the MLLM's ability to maintain and update its understanding of complex, unfolding events over time, complemented by a multi-dimensional evaluation structure.

# 5 Limitations and Future Work

Our benchmark reveals counter-intuitive findings, such as the limited impact of model scale and input frame count on performance, suggesting that MLLM mechanisms for processing continuous video are poorly understood and effective analysis tools are lacking. Furthermore, the current evaluation is primarily limited to the visual modality. Future work will focus on investigating the underlying causes of these phenomena and developing more adapted analytical methods. Concurrently, a key direction involves incorporating important modalities like audio into RTV-Bench to enable a more comprehensive evaluation of continuous perception, understanding, and reasoning in realistic multimodal scenarios.

# 6 Conclusion

In this work, we introduce **RTV-Bench**, a benchmark designed to systematically evaluate the continuous video understanding and real-time reasoning capabilities of multimodal large language models (MLLMs). RTV-Bench comprises 552 long-form and streaming-style videos paired with 4,608 carefully constructed QA instances, targeting time-varying perception, understanding, and reasoning under realistic online settings.

Extensive experiments across a diverse set of models reveal two key findings. First, models explicitly designed for online or streaming video processing consistently demonstrate stronger continuous understanding capabilities than general-purpose counterparts. Second, both parameter scaling and uniformly increasing the number of sampled frames during training or inference yield only limited and sometimes inconsistent performance gains. In particular, denser temporal sampling often leads to performance saturation or degradation, highlighting inherent limitations in current architectures when handling long or high-density visual streams. These observations underscore that effective real-time video understanding requires principled temporal modeling and selective information aggregation, rather than relying on naive increases in model size or frame counts.

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

# Appendix

The appendix includes the following sections:

- **Section A: Experimental Analysis Supplement.**
- **Section B: Framework Design and Application Extensions.**
- **Section C: Case.**

## A    Experimental Analysis Supplement

This section supplements the materials on Score Design and OAE Design, including related content and partial experimental data.

**Score VS Accuracy**    In Section 3.1, to comprehensively evaluate the performance of these models, we design two metrics: **Accuracy** for tack-specific correctness and **Score** for dynamic reasoning consistency. The Score metric is crucial for revealing reliable, hierarchical reasoning beyond simple Accuracy. While GPT-4o's Accuracy gain over IXC2.5-OL is moderate (50.02% vs. 47.33%), its substantially higher Score (22.10 vs. 15.40) indicates a more robust reasoning process, reliably building upon foundational understanding to address complex queries. Conversely, lower Scores relative to Accuracy, common among open-source models, suggest instability in multi-step reasoning. Thus, as shown in Figure 3, the Score metric effectively quantifies the deeper, more reliable comprehension capabilities demonstrated by models like GPT-4o on RTV-Bench's challenging tasks and it can distinguish model performance more clearly.

With the aim of validating the discriminative power of the Score metric, we conducted statistical analysis on 1,527 multi-timespan QA-triples, as shown in A.1. The result revealed a significantly positive correlation between foundational reasoning(Q1-Q2) and subsequent conplex reasoning performance(Q3). This demonstrates that models achieving higher accuracy on elementary visual perception tasks exhibit proportionally stronger performance on advanced temporal reasoning tasks.

Such alignment shows our core design philosophy for RTV-Bench: **hierarchical reasoning capabilities** depend on robust foundational perception, mirroring human cognitive processes in real-time video understanding. Furthermore, it also illustrates the rationality of **Score** as a real-time video metric, which addresses critical limitations of conventional single-answer accuracy metrics in assessing continuous reasoning dynamics.

Table A.1: Accuracy Distribution of Question 3 Conditioned on Preceding Question Performance: Q3 Accuracy when at least one of the preceding questions (Q1 or Q2) was answered correctly, versus Q3 accuracy when both preceding questions (Q1 and Q2) were answered incorrectly.

| Model | #Size | Q3 \| Q1/Q2≥1 | Q3 \| Q1&Q2=0 |
|---|---|---|---|
| GPT-4o [13] | – | 297 | 64 |
| IXC2.5-OL [47] | 7B | 499 | 81 |
| LLaVA-OneVision [15] | 7B | 309 | 182 |
| LLaVA-Video [22] | 7B | 308 | 195 |

**OAE Design Overview**    During evaluation, we also incorporated the Object-Action-Event framework( Table B.1) as part of our analytical scope, designed to assess video comprehension from multiple agent-centric perspectives. For example, in live sports scenarios, we evaluate three perspectives: objects (e.g., players appearing or disappearing during an offensive play), actions (e.g., dynamic maneuvers by offensive players), and events (e.g., real-time offensive strategies deployed in the midfield).

Table A.2: Detailed evaluation results on the category of **OAE**. Object, Action and Event.

| Method | #Size | Object Acc (%) / Score | Action Acc (%) / Score | Event Acc (%) / Score |
|---|---|---|---|---|
| *Open-Source Offline Video Models* | | | | |
| Qwen2.5-VL [43] | 7B | 39.67 / 8.1 | 38.12 / 6.7 | 37.18 / 6.8 |
| VideoLLaMA2 [4] | 7B | 40.39 / 8.2 | 40.25 / 8.6 | 38.69 / 6.8 |
| VideoLLaMA3 [45] | 7B | 34.31 / 4.5 | 37.77 / 7.4 | 34.21 / 3.9 |
| LLaVA-OneVision [15] | 7B | 34.31 / 5.0 | 35.82 / 4.9 | 33.42 / 3.4 |
| LLaVA-Video [22] | 7B | 36.10 / 6.1 | 35.40 / 4.7 | 33.88 / 3.8 |
| *Open-Source Online Models* | | | | |
| VITA-1.5 [9] | 7B | 47.39 / 13.7 | 44.09 / 11.6 | 42.85 / 10.3 |
| IXC2.5-OL [47] | 7B | 49.89 / 17.2 | 46.34 / 14.6 | 46.61 / 15.4 |
| *Closed-Source Business Models* | | | | |
| GPT-4o [13] | – | **50.63 / 23** | **50.97 / 22.3** | **49.01 / 20.9** |
| Gemini [35] 2.0 Flash | – | 42.66 / 12.2 | 43.34 / 11.5 | 40.24 / 12.4 |

Table A.3: Analytical Object Taxonomy: RTV-Bench systematically formulates an object-action-event framework, with explicit definitions of core characteristics and discriminative criteria for each category.

| Category | Dimensions |
|---|---|
| **Spatiotemporal Elements** | Objects: Physical entities appearing in video frames. Actions: Dynamic behaviors performed by objects. Events: Complex occurrences combining objects and actions. |

**OAE Accuracy and Score Analysis** In Table B.1, GPT-4o maintains the leading positions in all three dimensions of the OAE , and all online models demonstrate significantly higher accuracy and scores compared to offline models, particularly in the Object dimension, this highlights that the perspective design of OAE requires exceptionally strong capabilities in continuous analysis tasks. Overall, the models show no significant gaps in accuracy and scores across the three dimensions. Most offline models achieve their best performance in the Action aspect, while online models excel particularly in Object. Notably, nearly all models exhibit the lowest accuracy and scores in the Event dimension, indicating that continuous analysis tasks with higher complexity remain a formidable challenge for current systems.

## B    Framework Design and Application Extensions

This section supplements the materials on the rationale for and necessity of evaluation dimension design, while providing an extended analysis on the broader utility of the dataset.

**Methodological Foundations for Assessing Continuous Analysis Capabilities** To further elaborate on video continuous analysis capabilities introduced In Section 2.2, we formalize the foundational definitions and evaluation protocols for this capacity. Primarily, models are required to possess perception, understanding, and reasoning abilities comparable to state-of-the-art offline video models before addressing real-time streaming contexts. This requirement motivates our two-stage QA design, as elementary offline video analysis capabilities intuitively form the prerequisite for advanced temporal reasoning.

Furthermore, models must demonstrate proficiency in recognizing correct temporal query patterns due to three critical demands inherent to real-time applications: enhanced perception capabilities in highly dynamic scenarios requiring rapid and precise visual processing, deepened understanding of ongoing events under temporal continuity constraints, and effective reasoning about future trajectories based on evolving contextual cues.

**Why evaluate across perception, understanding, and reasoning dimensions?** To elucidate the necessity of three-dimensional categorization, we subsequently analyze perception, understanding, and reasoning respectively.In egocentric driving environments with high-dynamic scenarios, the video continuous analysis capability fundamentally addresses two aspects of persistent navigation: 1) holistic operational state awareness (e.g., current traffic condition assessment, historical route context) through the integration of offline and online video processing capabilities, and 2) perception speed, comprehension, and analytical capabilities for sudden real-time events.

Regarding perception design, this corresponds to detecting abrupt environmental changes during navigation, such as traffic light transitions, emergent vehicles, and pedestrians - scenarios where conventional video models exhibit critical deficiencies in temporal responsiveness. As visualized in Figure 4, existing architectures struggle to adapt to real-time variations in high-dynamic settings. We systematically decompose this capability into three sub-dimensions: Temporal Perception, Scene Perception and Visual Perception.

Regarding understanding design, it addresses the interpretative capacity for sudden operational changes during driving. For the understanding design, it pertains to scenarios involving the comprehension of abrupt changes during driving, such as interpreting traffic signal indications, road sign semantics, and the rationale behind preceding vehicles' maneuvering strategies. Previous models have been shown to fall short of fundamental requirements in both processing speed and interpre-

Table B.1: Core Evaluation Dimensions: RTV-Bench systematically defines eight essential evaluation dimensions for continuous video understanding systems, accompanied by formal characterizations and discriminative criteria for each dimension.

| Category | Dimensions |
|---|---|
| **Model Capabilities** | Temporal Perception (TP): Recognizing temporal sequence and duration. Scene Perception (SP): Understanding holistic environment and layout. Visual Perception (VP): Detecting fine-grained visual features. Future Prediction (FP): Anticipating future developments. Phenomenological Understanding (PU): Interpreting surface phenomena. Intent Analysis (IA): Inferring actor motivations. Global Understanding (GU): Grasping video context. Spatiotemporal Reasoning (SR): Logical deduction from observations. |

tative depth. For instance, during traffic signal transitions or lane-changing events, current models struggle to detect such changes within reasonable timeframes. In complex real-time traffic scenarios, beyond insufficient processing speed, existing systems also largely fail to meet advanced comprehension requirements for situational awareness. Through systematic categorization and abstraction of diverse scenarios, we decompose this dimension into three sub-components: Intent Analysis, Phenomenological Understanding and Global Understanding.

Regarding reasoning design, it encompasses scenarios requiring continuous temporal analysis, such as traffic signal duration estimation, and anticipating preceding vehicles' strategies, these demand sophisticated continuous analysis capabilities. Based on the dual requirements of historical context integration and prospective forecasting, we architect this dimension into two sub-dimensions: Spatiotemporal Reasoning and Future Prediction.

**Why cross-apply the OAE design with eight evaluation dimensions?**   We observe notable limitations in the dimensional design frameworks of current mainstream benchmarks, suggesting areas that warrant systematic refinement. Current evaluation taxonomies frequently include components such as Object Recognition and Action Reasoning, yet conspicuously omit complementary dimensions like Object Reasoning and Action Recognition. This prevalent pattern reveals significant arbitrariness in dimension partitioning, resulting in evaluation frameworks whose systematic rigor and scientific credibility remain fundamentally compromised. To address these limitations, we propose a novel taxonomy that first independently categorizes analytical subjects and evaluation dimensions, subsequently implementing cross-categorization to establish a structured and systematic grid framework for dimensional organization.

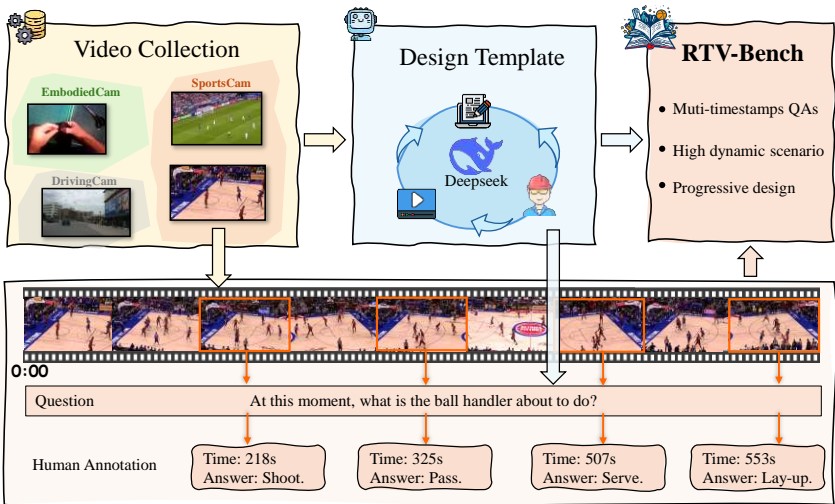

Figure B.1: Our dataset construction pipeline. We develop a dataset generation pipeline consisting of three stages to create RTV-Bench: Video Collection, Templates Design and Human Annotation.

**Supplementary Analysis of Dataset Value**   This work introduces the Real-time Video Understanding Benchmark (RTV-Bench), which holds significant potential to advance video comprehension research and its applications. By defining core capabilities required for real-time video processing through high-dynamic video evaluation, RTV-Bench establishes continuous analysis capabilities as critical developmental objectives. The benchmark demonstrates a novel paradigm for online video capability assessment, potentially accelerating the enhancement of core functionalities in online video understanding models and expediting their deployment in real-time scenarios such as embodied AI, autonomous driving, and live sports commentary.

Experimental analyses using RTV-Bench reveal substantial deficiencies in existing video models' continuous analysis capabilities while illuminating potential improvement pathways. Our results indicate that increasing model scale or video sampling frequency yields diminishing returns. The substantial performance gap between online and offline models in continuous analysis capabilities not only validates RTV-Bench's design efficacy but also suggests architectural innovation as a viable development direction.

The RTV-Bench framework is constructed upon three foundational principles: 1) Balanced Taxonomy Design: Prioritizing structural rationality in evaluation dimensions while implementing novel task formulations. 2) QA Integrity Assurance: Maintaining high-quality question-answer pairs while mitigating validity threats from oversimplified QA designs, particularly accuracy inflation through correct guessing in multiple-choice formats. 3) Capability Anchoring: Conducting specialized evaluations for real-time video processing while systematically monitoring potential degradation in foundational video understanding competencies. Furthermore, the rigorous RTV-Bench generation pipeline B.1 also ensures high-quality QA pair production B.2.

**Broader Impact**   By focusing on real-time video comprehension, RTV-Bench rigorously evaluates models' continuous temporal modeling and dynamic scenario comprehension capabilities. This benchmarking framework directly addresses critical gaps in deploying video understanding systems for latency-sensitive applications, including embodied AI navigation requiring sub-second environmental feedback, autonomous driving systems dependent on frame-by-frame hazard anticipation, and live sports commentary generation demanding real-time event contextualization.

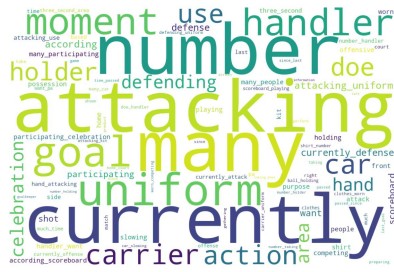

Figure B.2: This figure presents a word cloud visualization derived from QA question corpora, explicitly highlighting the most frequently occurring terms through typographic prominence.

Regarding potential negative societal impact, while RTV-Bench can better facilitate the development and application of real-time video tasks, these technologies may concurrently promote the creation of more sophisticated real-time surveillance systems, which inherently raises privacy concerns. Therefore, the advancement of related technologies must be conducted under the supervision of robust privacy protection measures and ethical standards.

## C   Case

We present additional examples from RTV-Bench, covering eight evaluation dimensions and three analytical perspectives, to demonstrate the holistic and systematic framework for assessing continuous analysis capabilities.

**Temporal Perception**

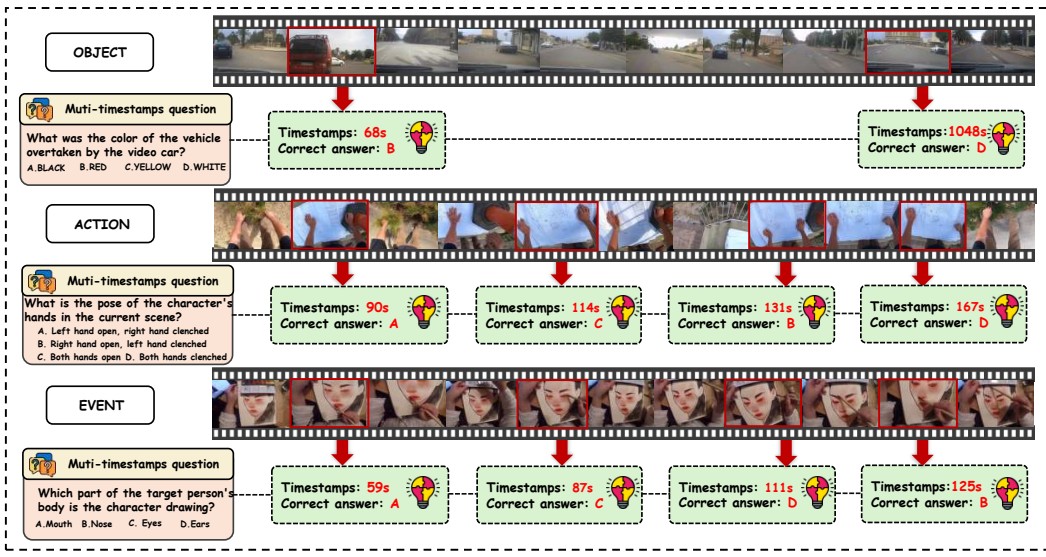

**Scene Perception**

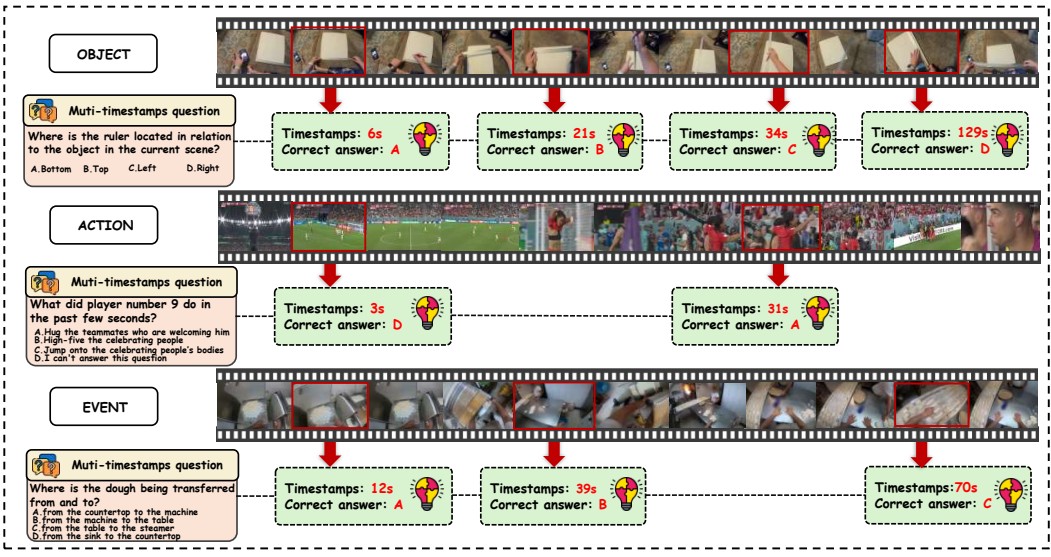

## Visual Perception

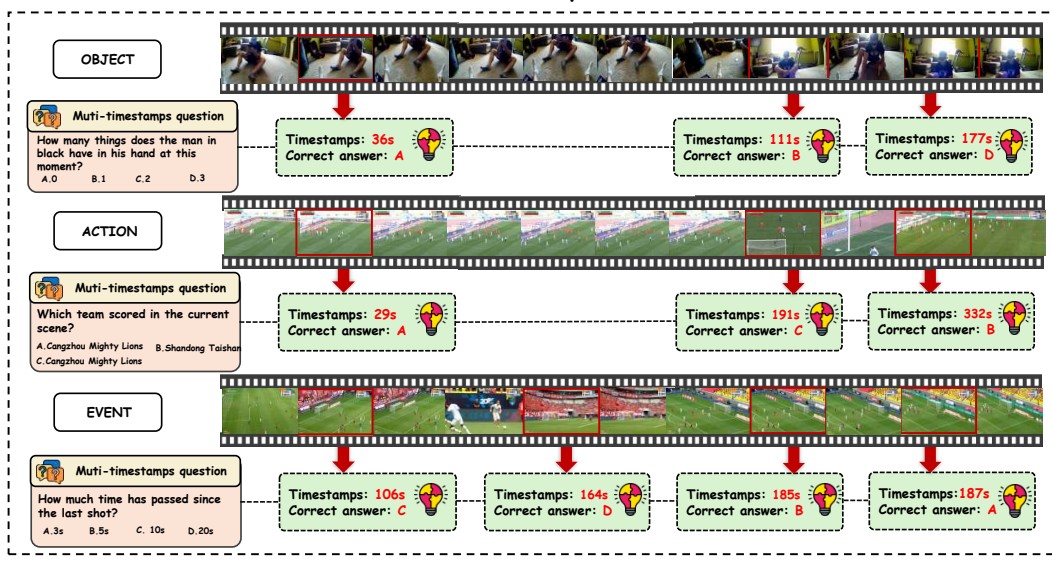

## Spatiotemporal Reasoning

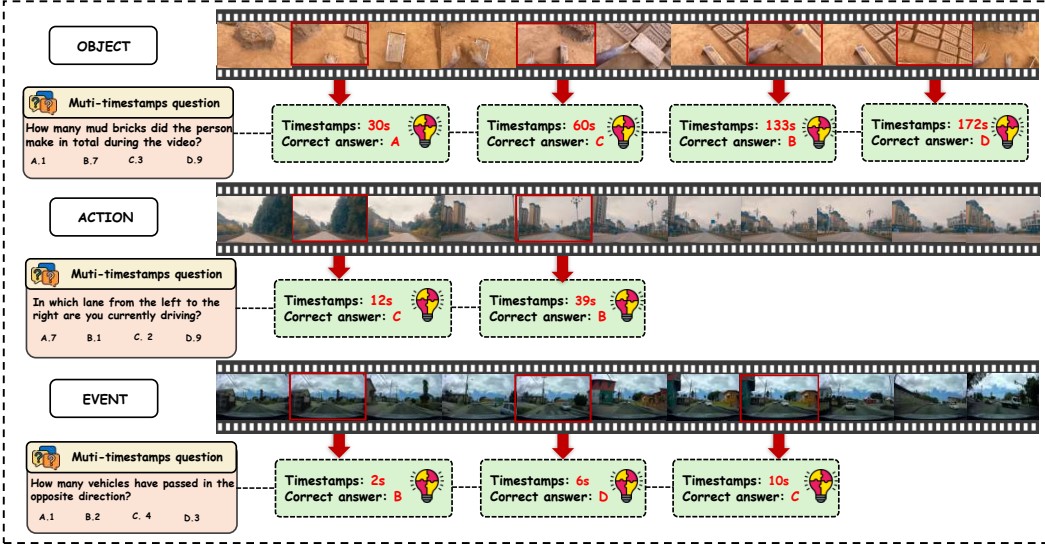

## Global Understanding

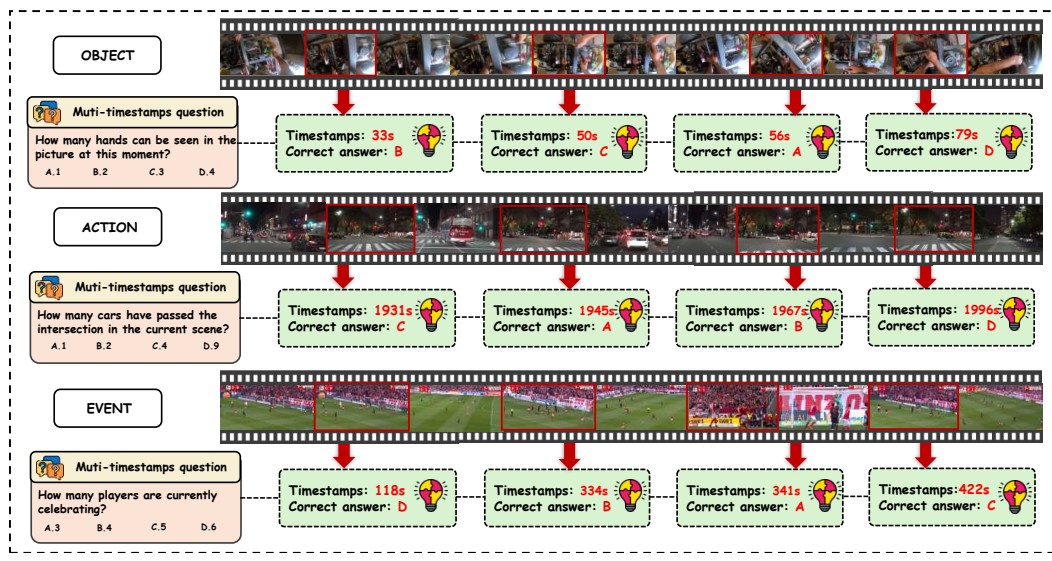

## Intent Analysis

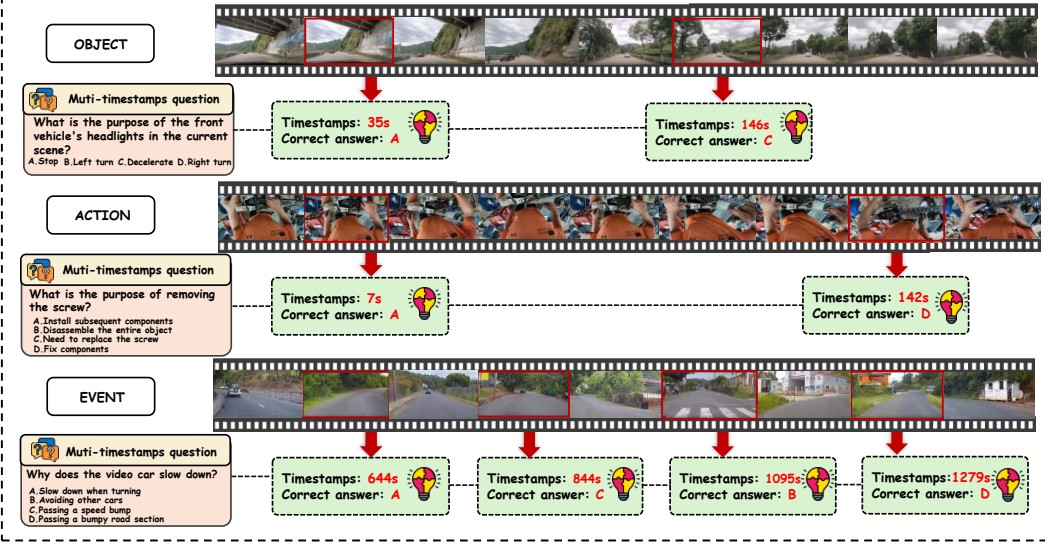

## Phenomenological Understanding

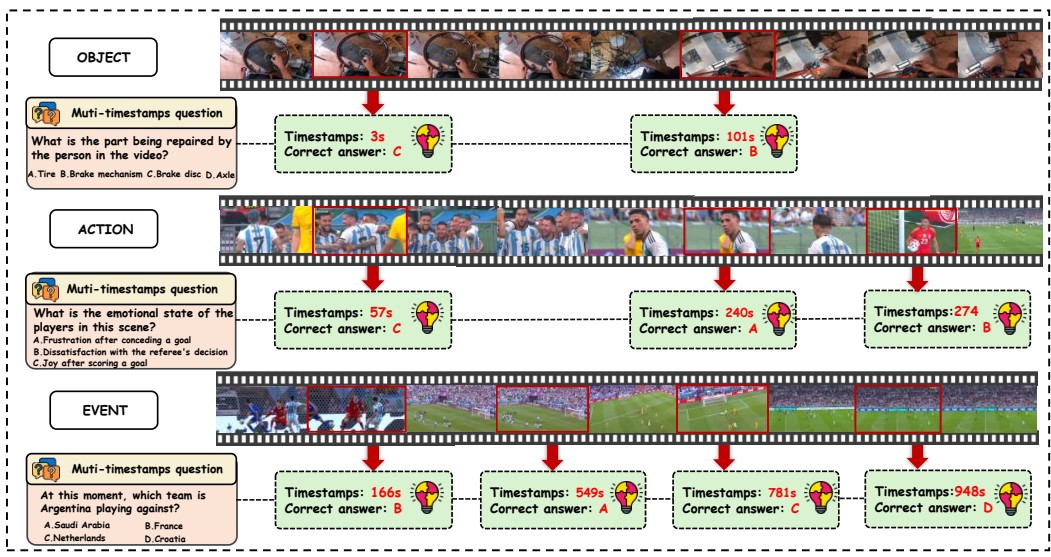

## Future Prediction

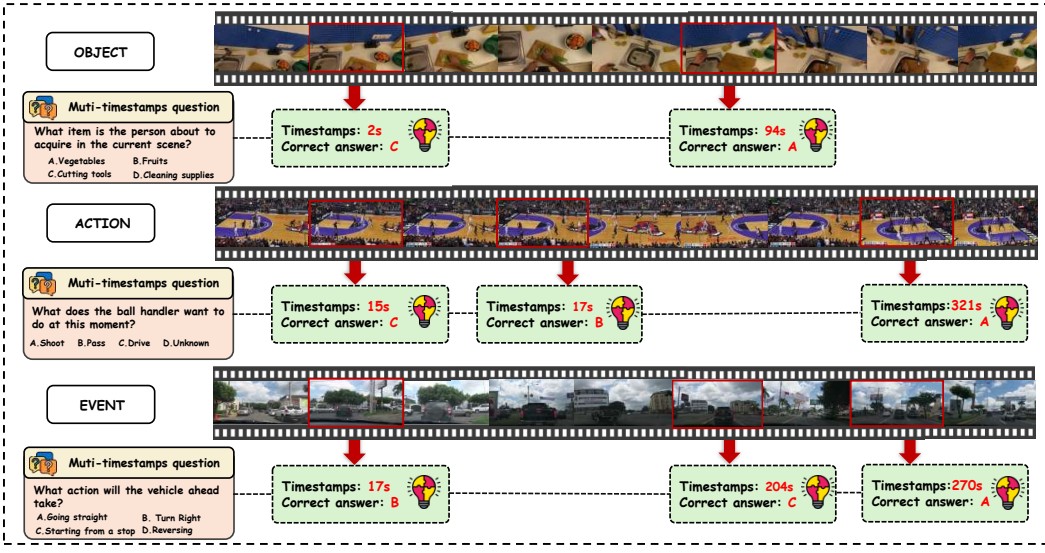

