# OpenReview forum: "RTV-Bench: Benchmarking MLLM Continuous Perception, Understanding and Reasoning through Real-Time Video"
_NeurIPS.cc/2025/Datasets_and_Benchmarks_Track — NeurIPS 2025 Datasets and Benchmarks Track poster_

### Official Review · Reviewer_sDWp · 2025-06-23

**Rating:** 5
**Confidence:** 3

**Summary:**

This paper introduces RTV-Bench, a new benchmark designed to evaluate the real-time responsiveness of VideoLLMs. RTV-Bench consists of three key components aimed at assessing continuous video analysis capabilities: (1) Multi-Timestamp Q&A with questions whose answers evolve within a video, (2) Hierarchical Question Structure enforcing reliable, sequential reasoning, and (3) Multi-dimensional Evaluation for providing granular diagnostics. Using this benchmark, the authors conduct a comprehensive empirical analysis and report several key findings: (1) most existing VideoLLMs achieve less than 50% accuracy, (2) performance exhibits weak correlation with either the number of input frames or model size, and (3) models explicitly tailored for real-time video understanding significantly outperform those designed for offline video settings.

**Additional Feedback:**

Overall, the proposed benchmark is well-motivated and addresses an important gap in evaluating real-time video understanding, with the potential to inspire future research in the field. I encourage the authors to address the concerns outlined in the weaknesses section to further strengthen the contribution.

**Dataset Code Accessibility:**

Partly

**Dataset Code Comments:**

The authors have released the dataset, but there isn't sufficient detail for the reproduction of results in the document.

**Ethical Considerations:**

No, there are no or only very minor ethics concerns

**Final Justification:**

Thank you to the authors for their response. While several of my concerns have been adequately addressed, some key experiments remain unprovided. I encourage the authors to include a more detailed description of the QA annotation pipeline and to conduct additional experiments, particularly with blind language models and models' behaviors under diverse scenarios. Given these, I am inclined to maintain my original rating.

**Limitations Weaknesses:**

1. **Insufficient detail on QA annotation process**: The paper lacks sufficient information regarding the construction of QA annotations, particularly for advanced-level questions. Providing more details on the human annotation pipeline would help assess the quality and consistency of the dataset.

2. **Limited analysis of model behavior**: The empirical analysis would benefit from a more fine-grained examination of model behaviors. For instance, it remains unclear how models respond to the same question at different timestamps, or how their performance varies across question difficulty levels. Additionally, evaluating a blind language model (i.e., without video input) could serve as a useful baseline to quantify the reliance of current models on visual context.

**Strengths Contributions:**

1. **Well-motivated benchmark design**: The proposed benchmark is clearly motivated by practical needs in real-world video understanding scenarios, specifically targeting real-time responsiveness, an underexplored but critical aspect. This focus distinguishes it meaningfully from existing benchmarks that predominantly emphasize offline settings.

2. **High-quality human annotations**: In contrast to prior benchmarks that often rely heavily on machine-generated annotations (e.g., from large language models), the authors mainly employ human annotators for dataset construction. This likely results in higher annotation fidelity and reliability.

3. **Insightful empirical findings**: The analysis reveals several counter-intuitive observations, such as the weak correlation between model size or number of input frames and benchmark performance. These findings may inspire future research on designing more efficient and effective VideoLLMs.

---

> ### Author Rebuttal · Authors · 2025-07-31
>
> Thank you for your valuable suggestions. We hope our response has addressed your concerns.
>
> > **w1**: The paper lacks sufficient information regarding the construction of QA annotations, particularly for advanced-level questions. Providing more details on the human annotation pipeline would help assess the quality and consistency of the dataset.
>
> **Ans**: We have provided a brief description of the QA annotation process in both the main text (Section 2.3 Manual Annotation) and the appendix (appendix.pdf, Figure B.1). The specific workflow is as follows: manually select videos from three categories - EmbodiedCam, SportsCam, and DrivingCam; design templates by combining human design with AI assistance; then conduct annotation in groups on our self-developed annotation platform; finally perform screening and AI verification after completion.
>
> > **w2**:The empirical analysis would benefit from a more fine-grained examination of model behaviors. For instance, it remains unclear how models respond to the same question at different timestamps, or how their performance varies across question difficulty levels. Additionally, evaluating a blind language model (i.e., without video input) could serve as a useful baseline to quantify the reliance of current models on visual context.
>
> **Ans**: We appreciate the reviewer’s suggestion. We agree that finer-grained analyses, such as model responses to the same question at different timestamps and across difficulty levels, are valuable. Evaluating a blind language model baseline is also important; Dang et al. [1] show that blind models perform poorly in video understanding, losing almost all visual comprehension. Due to space limits, these experiments will be conducted in future work.
>
> > about **Dataset Code Comments & Additional Feedback**:The authors have released the dataset, but there isn't sufficient detail for the reproduction of results in the document.
>
> **Ans**: We sincerely appreciate your attention to the reproducibility of our dataset results. In the supplementary materials submitted with our paper, we have included our code implementation and made a commitment to open-source the complete code for community use.
>
> ---
> [1] Dang Y, Gao M, Yan Y, et al. Exploring response uncertainty in mllms: An  empirical evaluation under misleading scenarios[J]. arXiv preprint  arXiv:2411.02708, 2024.

---

> > ### Comment · Reviewer_sDWp · 2025-08-02
> >
> > Thank you to the authors for their response. While several of my concerns have been adequately addressed, some key experiments remain unprovided. I encourage the authors to include a more detailed description of the QA annotation pipeline and to conduct additional experiments, particularly with blind language models and models' behaviors under diverse scenarios. Given these, I am inclined to maintain my original rating.

---

> > > ### Author Response · Authors · 2025-08-09
> > >
> > > Thank you for your valuable comments and guidance on our work, which have been very helpful for improving our paper. We will consider supplementing relevant content in our future work. Finally, we would like to thank you again for your recognition of our work.

---

### Official Review · Reviewer_8LUb · 2025-07-01

**Rating:** 4
**Confidence:** 5

**Summary:**

The paper presents a new dataset for real-time video question answering and is focused on a practical setting: continuous question answering, where the answer to a question is dynamic and changeable through the video. The paper has also benchmarked a list of offline and online methods regarding the QA accuracy.

**Additional Feedback:**

I would appreciate seeing the justifications or complementary results mentioned in the weaknesses section. Overall, the multi-timestamp or time-varying question answering setting is a valuable contribution to the broader landscape of online VideoQA. However, the analysis and writing would benefit from further refinement to better distinguish this work from prior studies and to enhance the clarity and utility of the benchmark for the research community.

My initial rating is borderline, and my final decision will depend on whether the rebuttal can resolve my concerns.

**Dataset Code Accessibility:**

Yes

**Ethical Considerations:**

No, there are no or only very minor ethics concerns

**Final Justification:**

Thanks for the detailed response. While some of my concerns have been addressed, I believe the key issue, the emphasis on the time-evolving nature of the setting, remains understated. I acknowledge that this partly stems from a misalignment in the presentation that couldn’t be fully resolved in the rebuttals. Regarding w4, I would like to clarify that I didn’t imply latency in terms of seconds, as I agree the paper isn’t focused on inference efficiency. Rather, I was referring to temporal delays across timesteps or frame indices, which could be described as a ‘semantic delay.’ Thus, I’ve decided not to raise my rating.

**Limitations Weaknesses:**

1. Missing baselines: a few relevant baseline models are missing, which is conducive to the comprehensiveness of the dataset: such as VideoLLM-online (CVPR24), Flash-VStream, etc.

2. Missing metrics: the *score* reflects the accuracy of advanced questions when the basic questions are correctly answered. I think it is necessary to show the absolute value of the accuracy on $q_0$, $q_1$, $q_2$, since it is also interesting to see the ratio of the correctly straightforwardly answered $q_2$ questions, even when the basic questions are incorrectly answered. This is a measure of the question quality and the question difficulty.

3. It is interesting and necessary to show how/if the LLM responses evolve as the video unfolds. This is a particularly pertinent discussion, as the paper focuses on continuous/time-varying reasoning. Some intuitive questions arise:
- Are the online models biased or largely dependent on the initial perception?
- Are there any temporal delays from the model response as the video unfolds?
- The paper shows the distribution of query time in Fig. 2; is the model performance significantly worse on the *deep* queries than the *shallow* queries?


4. The paper is easy to follow, but the presentation still needs polishing:
- Task taxonomy (perception/reasoning/understanding) and categories (e.g., Temporal Perception, Intent Analysis, etc.) need formal definitions and clarification in the scope of the paper. Even though a few terms are widely used in the community, a formal and easy-to-find definition is necessary for readers' reference.
- Personally, I think '*continuous*' is a broad and vague word in this work. The paper highlights the multi-timestamp QA and time-varying answers, and *continuous* fails to convey the message clearly and specifically.

**Strengths Contributions:**

1. Motivation: The motivation is well-grounded. To the best of my knowledge, the problem of dynamic answers in the streaming/online video setting remains unexplored, despite the growing interest in streaming video QA. Addressing this gap makes the proposed dataset timely and meaningful.

2. Dataset Design: The dataset incorporates fine-grained categories and a hierarchical question structure. This design supports a more nuanced evaluation of follow-up questions and questions of different difficulties.

2. The paper conducts an inclusive comparison across a wide range of models, covering both online and offline settings.

---

> ### Author Rebuttal · Authors · 2025-07-31
>
> We sincerely appreciate your valuable feedback on our experimental analysis and writing. Each of your suggestions has been immensely beneficial to us.
> > **w1**: Missing baselines: a few relevant baseline models are missing, which is conducive to the comprehensiveness of the dataset.
>
> **Ans**: We appreciate the reviewer’s suggestion regarding additional baselines. Owing to time limitations, we could not include them in this version, but we will incorporate them in future updates to improve the dataset’s comprehensiveness.
>
> > **w2**: Missing metrics: the score reflects the accuracy of advanced questions when the basic questions are correctly answered. I think it is necessary to show the absolute value of the accuracy on q0, q1, q2, since it is also interesting to see the ratio of the correctly straightforwardly answered questions.
>
> **Ans**: Thank you for your insightful comments on the experimental analysis. Your suggestions have significantly deepened our understanding. Regarding the analysis of basic (q0/q1) and advanced (q2) questions, we provide the absolute accuracy values and ratios in the table below. Although some of this data was included in the main text and appendix, we did not conduct an in-depth analysis initially. Under your guidance, we further analyzed the results and found that for offline models, the accuracy of basic questions and q2 questions is nearly identical. However, for online models, the performance on basic questions improves significantly, while q2 accuracy sees only a modest increase.
>
> This suggests that improvements in real-time capability primarily stem from enhanced foundational perception and comprehension abilities, which also partially explains the superior performance of some commercial closed-source models.
> | Method           | q0&q1 acc | q2 acc | q0&q1 absolut value | q2 absolut value |
> |------------------|-----------|--------|---------------------|------------------|
> | Qwen2.5-VL       | 31.37     | 34.37  | 645/2056            | 877/2552         |
> | VideoLLaMA2      | 45.77     | 34.95  | 941/2056            | 892/2552         |
> | VideoLLaMA3      | 36.43     | 34.84  | 749/2056            | 889/2552         |
> | LLaVA-OneVision  | 35.80     | 33.58  | 736/2056            | 857/2552         |
> | VITA-1.5         | 55.06     | 36.32  | 1132/2056           | 927/2552         |
> | IXC2.5-OL        | 59.05     | 38.21  | 1214/2056           | 975/2552         |
> | GPT-4o           | 56.53     | 44.73  | 1162/2056           | 1142/2552        |
> | Gemini 2.0 Flash | 47.49     | 38.64  | 976/2056            | 986/2552         |
>
> > **w3**: Are the online models biased or largely dependent on the initial perception?
>
> **Ans**: First, we should clarify that our q0/q1 questions are designed to assess what you refer to as "initial perception," primarily involving single-frame question-answering before the q2 questions. Following this logic, we can extend the discussion from the previous response. The significant improvement in basic question accuracy for online models, compared to the modest increase in q2 performance, highlights the critical role of initial perception in enabling subsequent reasoning capabilities.
>
> > **w4**: Are there any temporal delays from the model response as the video unfolds?
>
> **Ans**: We appreciate the reviewer’s insightful question regarding potential temporal delays in model responses. Temporal alignment is indeed a key challenge in real-time video understanding. While our current focus lies in improving the model's comprehension over video streams, rather than in minimizing response latency per se, we acknowledge the importance of evaluating and potentially reducing such delays. We will consider this dimension more explicitly in future work.
>
> > **w5**: The paper shows the distribution of query time in Fig. 2; is the model performance significantly worse on the deep queries than the shallow queries?
>
> **Ans**: We summarize the statistics in the table below. It is evident that as questions become deeper, model accuracy gradually declines, though the drop is relatively modest.
> | model/time(s) | (0,30] | (30,60] | (60,180] | (180.300] | (300,600] | (600,1800] | (1800,3600] | 3600+ |
> | --- | --- | --- | --- | --- | --- | --- | --- | --- |
> | Qwen | 43.07 | 32.97 | 32.40 | 28.69 | 32.28 | 29.85 | 25.58 | 25.00 |
> | IXC | 52.02 | 47.83 | 47.12 | 40.95 | 40.76 | 47.70 | 43.79 | 39.53 |
> | VideoLLama3 | 45.60 | 36.93 | 37.52 | 32.21 | 32.47 | 31.06 | 31.68 | 30.63 |
> | VideoLLaMA2-7B | 44.77 | 43.28 | 41.11 | 36.57 | 39.40 | 37.47 | 40.12 | 29.35 |
> | Llava | 34.31 | 35.73 | 35.81 | 33.74 | 38.58 | 33.77 | 33.72 | 33.70 |
> | Gpt | 61.22 | 49.31 | 50.48 | 47.60 | 50.76 | 43.93 | 44.29 | 38.46 |
> | Gemini | 52.78 | 45.47 | 40.61 | 36.73 | 38.52 | 38.14 | 35.68 | 35.00 |
>
> > **w6**：Personally, I think 'continuous' is a broad and vague word in this work. The paper highlights the multi-timestamp QA and time-varying answers, and continuous fails to convey the message clearly and specifically.
>
> **Ans**: Regarding the wording of "continuous," let us first clarify how our work differs from datasets like OVO[1], StreamingBench[2], and MMBench[3]. Our focus is on the high-dynamic nature of online models. Taking perception as an example, traditional perception aligns more closely with our q0/q1 tasks—essentially indistinguishable from single-frame image perception (which we argue is fundamentally the same). In contrast, online models require continuous perception, emphasizing the ability to detect changes in highly dynamic scenes.
>
>
> We define this continuous perception and comprehension ability as follows:
> 1）Prioritizing recent changes over long-term historical data (e.g., sudden road hazards during prolonged driving).
> 2）Dynamic contextual recall—retrieving relevant information from memory (e.g., recalling a road sign seen earlier or remembering where an object was placed).
> 3）Spatiotemporal integration—understanding past, present, and future states in a temporal stream (e.g., predicting a leading vehicle's next move and adjusting driving strategy accordingly).
> All these occur in a continuous scenario, testing the model's ability to perceive and comprehend abrupt changes. Hence, we refer to it as "continuous understanding" rather than mere "understanding," as it represents a higher-dimensional capability built upon high-dynamic online video processing.
>
> > **w7**:Task taxonomy (perception/reasoning/understanding) and categories (e.g., Temporal Perception, Intent Analysis, etc.) need formal definitions and clarification in the scope of the paper. Even though a few terms are widely used in the community, a formal and easy-to-find definition is necessary for readers' reference.
>
> **Ans**: The task taxonomy and categories are briefly described and defined in the Appendix (appendix.pdf, Table B.1). First, let me explain the task taxonomy (using drivecam videos as an example):
>
> **Perception** is designed to test the online model's ability to perceive highly dynamic events (i.e., whether it can promptly detect changes, such as traffic light transitions, sudden vehicles, or pedestrians).
>
> **Understanding** (inspired by OVO's "Real-time understanding") evaluates the online model's comprehension of highly dynamic events (i.e., whether it can promptly interpret them, such as understanding traffic signal instructions, parsing road sign semantics, or analyzing the maneuvering logic of leading vehicles).
>
> **Reasoning** assesses the online model's capability for complex, continuous analysis of highly dynamic events (i.e., whether it can sustain predictions and decision-making, such as forecasting traffic light cycles).
> Next, we define and explain the specific categories:
>
> For **Perception** categories:Temporal Perception/Scene Perception/Visual Perception are evaluated from three dimensions: short-duration dynamic events, scene changes, and minor visual variations, respectively, forming the types of perception.
>
> For **Understanding** categories:Intent Analysis (referencing VideoMind and OVO)/Phenomenological Understanding/Global Understanding evaluate three dimensions: motivation analysis, phenomenon understanding, and spatial comprehension capabilities, respectively.
>
> For **Reasoning** categories:Future Prediction (OVO)/Spatiotemporal Reasoning evaluate two dimensions: future prediction and temporal reasoning capabilities.
>
> Overall, this forms a progressive relationship, while the categories represent parallel multi-dimensional evaluations.
> The entire testing logic is conducted in a highly dynamic, real-time video context, which is derived from the three perspectives of continuous perception and understanding capabilities mentioned above (w6).
>
> ---
>
> [1] Niu J, Li Y, Miao Z, et al. OVO-Bench: How Far is Your Video-LLMs from Real-World Online Video Understanding?[C]//Proceedings of the Computer Vision and Pattern Recognition Conference. 2025: 18902-18913.
> [2] Lin J, Fang Z, Chen C, et al. Streamingbench: Assessing the gap for mllms to achieve streaming video understanding[J]. arXiv preprint arXiv:2411.03628, 2024.
> [3] Fang X, Mao K, Duan H, et al. Mmbench-video: A long-form multi-shot benchmark for holistic video understanding[J]. Advances in Neural Information Processing Systems, 2024, 37: 89098-89124.

---

> > ### Comment · Reviewer_8LUb · 2025-08-05
> >
> > Thanks for the detailed response. While some of my concerns have been addressed, I believe the key issue, the emphasis on the time-evolving nature of the setting, remains understated. I acknowledge that this partly stems from a misalignment in the presentation that couldn’t be fully resolved in the rebuttals. Regarding w4, I would like to clarify that I didn’t imply latency in terms of seconds, as I agree the paper isn’t focused on inference efficiency. Rather, I was referring to temporal delays across timesteps or frame indices, which could be described as a ‘semantic delay.’ Thus, I’ve decided not to raise my rating.

---

> > > ### Author Response · Authors · 2025-08-07
> > >
> > > We apologize for the misunderstanding and sincerely thank you for your continued engagement with our work. Upon further reflection on your comments regarding the **time-evolving nature of the setting**, we would like to take this opportunity to clarify our intention.
> > >
> > > In **Figure 4**, we present case studies comparing model responses to the same question at different time points in the same video, aiming to highlight biases that arise during temporal evolution. **Appendix C** further provides more detailed examples to illustrate how current video models behave over time. Specifically, our benchmark includes a deliberate design where the same question is asked at different moments within a video to examine how model responses change—posing a significant challenge for existing models.
> > >
> > > Your comments made us realize that we could introduce even more fine-grained metrics to capture temporal dynamics more explicitly. While we agree that there is room for improvement, we believe our current efforts represent a strong first step toward evaluating video models under streaming and temporally-evolving conditions. We plan to explore more comprehensive designs in future work.

---

### Official Review · Reviewer_G4vi · 2025-07-03

**Rating:** 4
**Confidence:** 4

**Summary:**

This paper introduces RTV-Bench, a benchmark for evaluating continuous perception, understanding, and reasoning of multimodal large language models (MLLMs) in real-time video analysis. The benchmark incorporates multi-timestamp QA, hierarchical question structures, and multi-dimensional evaluation, with 552 videos and 4,631 QA pairs. Experiments show real-time models outperform offline ones, but proprietary models (e.g., GPT-4o) still lead.

**Dataset Code Accessibility:**

Yes

**Ethical Considerations:**

No, there are no or only very minor ethics concerns

**Final Justification:**

Based on feedback and opinions from other reviewers, I have decided to maintain my initial score.

**Limitations Weaknesses:**

- While Multi-Timestamp QA (MTQA) evaluates dynamic responses, the benchmark lacks explicit testing of long-term temporal dependencies. For instance, questions requiring models to link events across minutes (e.g., "How did the team strategy evolve from the first half to the second?") are absent, limiting the assessment of sustained context retention.

- Although hierarchical questions are included, the final evaluation still relies on the correctness of the answers. For questions involving complex reasoning (such as causal chains and counterfactual reasoning), merely judging whether the answers are right or wrong may not fully capture the depth and robustness of the model's reasoning process. More sophisticated evaluation metrics, such as scoring reasoning steps, analyzing error types, will provide more comprehensive insights.

**Strengths Contributions:**

- RTV-Bench addresses the gap in evaluating continuous analysis for MLLMs by introducing dynamic timestamp-based QA, hierarchical reasoning tasks, and fine-grained evaluation dimensions (e.g., Temporal Perception, Spatiotemporal Reasoning), which surpasses static benchmarks.

- The study covers diverse models (proprietary, open-source offline/online) and uses two metrics (Accuracy and Score) to assess both task correctness and hierarchical reasoning consistency, providing robust insights into model capabilities.

- The findings reveal that real-time models outperform offline counterparts but struggle with complex reasoning, and that model scale/frame rates do not correlate with performance.

---

> ### Author Rebuttal · Authors · 2025-07-31
>
> > **w1**: While Multi-Timestamp QA (MTQA) evaluates dynamic responses, the benchmark lacks explicit testing of long-term temporal dependencies. For instance, questions requiring models to link events across minutes (e.g., "How did the team strategy evolve from the first half to the second?") are absent, limiting the assessment of sustained context retention.
>
> **Ans**: We sincerely appreciate your valuable insights. Regarding the absence of long-term temporal dependency evaluation, we have actually incorporated relevant designs, such as counting-based templates (e.g., "How many... have appeared in the video up to now?") and state transition templates (e.g., "In the current scene, where is the object moving from and to?"), to assess long-term temporal dependencies.
>
> > **w2**: Although hierarchical questions are included, the final evaluation still relies on the correctness of the answers. More sophisticated evaluation metrics, such as scoring reasoning steps, analyzing error types, will provide more comprehensive insights.
>
> **Ans**: We thank the reviewer for their constructive suggestions on evaluation metrics. While our current assessment primarily depends on answer correctness, we fully agree that introducing more nuanced evaluation methods—such as scoring reasoning steps or analyzing error types—would provide deeper insights into model performance. We will prioritize this direction as a key area for future improvement.

---

> > ### Comment · Reviewer_G4vi · 2025-08-01
> >
> > Thank you for the response.
> >
> > Scoring reasoning steps or analyzing error types is indeed a valuable yet challenging task. I would recommend considering the designs in [1] [2], which convert the evaluation of reasoning processes into validation of answer correctness by introducing multi-level and multi-granularity QA. I believe it would be beneficial for the authors to incorporate this discussion into the paper.
> >
> > Based on feedback and opinions from other reviewers, I have decided to maintain my initial score.
> >
> > [1] Li X, Li X, Hu S, et al. CausalStep: A Benchmark for Explicit Stepwise Causal Reasoning in Videos[J]. arXiv preprint arXiv:2507.16878, 2025.
> > [2] Hu S, Zhang D, Feng X, et al. A multi-modal global instance tracking benchmark (mgit): Better locating target in complex spatio-temporal and causal relationship[J]. Advances in Neural Information Processing Systems, 2023, 36: 25007-25030.

---

> > > ### Author Response · Authors · 2025-08-07
> > >
> > > Thank you for your thoughtful comments and suggestions. We appreciate your recommendation of [1] and [2], and we agree that incorporating multi-level, fine-grained QA is a promising direction. We will include a discussion of these works in the revised version.
> > >
> > > [1] Li X, Li X, Hu S, et al. CausalStep: A Benchmark for Explicit Stepwise Causal Reasoning in Videos[J]. arXiv preprint arXiv:2507.16878, 2025.
> > >
> > > [2] Hu S, Zhang D, Feng X, et al. A multi-modal global instance tracking benchmark (mgit): Better locating target in complex spatio-temporal and causal relationship[J]. Advances in Neural Information Processing Systems, 2023, 36: 25007-25030.

---

> ### Author Response · Authors · 2025-08-01
>
> Thank you for your insightful and constructive suggestion. We agree that **scoring reasoning steps** and analyzing error types are both valuable and challenging directions.
>
> We will incorporate the discussion on evaluating reasoning processes into our paper, particularly drawing inspiration from the multi-level and multi-granularity QA designs proposed in [1][2]. We believe this addition will offer a fresh perspective and practical foundation for assessing the real-time reasoning capabilities of video models.
>
> ----
> [1] Li X, Li X, Hu S, et al. CausalStep: A Benchmark for Explicit Stepwise Causal Reasoning in Videos[J]. arXiv preprint arXiv:2507.16878, 2025.
>
> [2] Hu S, Zhang D, Feng X, et al. A multi-modal global instance tracking benchmark (mgit): Better locating target in complex spatio-temporal and causal relationship[J]. Advances in Neural Information Processing Systems, 2023, 36: 25007-25030.

---

### Official Review · Reviewer_SZ3Q · 2025-07-03

**Ethics Flags:** Data privacy, copyright, and consent
**Rating:** 5
**Confidence:** 5

**Summary:**

This paper proposes a novel dataset focusing on accessing the online understanding ability of models. The authors collects the videos from EgoSchema and online sources, and filters long videos with high dynamics and real-time needs. These videos are mannually annotated with questions and answers. Such annotations are also manually verified.  Exepriments on this benchmark reveal the limitations of current SOTA Video-MLLMs on stream video understanding and long sequences.

**Additional Feedback:**

Generally, the reviewer likes the paper and apprecites the authors efforts on building such benchmark. However, the authors are suggested to upload an ungated version of the dataset (possibly on another platform) for the reviewer to evaluate the reproducibility of the benchmark. With such limitation on reproducibility, the reviewer can only rate borderline accept rather than higher score on this stage.

**Dataset Code Accessibility:**

No

**Dataset Code Comments:**

The dataset is hosted on Huggingface. However, the dataset is gated and accessing this dataset requires sharing the contact information (email and huggingface username) with the repository authors, which may break the single-blind revewing process.

**Ethical Comments:**

The authors claim that part of videos are collected form online sources, but no explicit source is described. Whether the usage of these videos aligns with the user policy of such sources needs further verification.

**Ethical Considerations:**

Yes, there are ethics concerns that require attention by the authors

**Final Justification:**

I have read the authors' rebuttal with reviews and discussions from peer reviewers. I believe most of my concerns have been addressed. Therefore, I recommand acceptance for this paper.

**Limitations Weaknesses:**

- [Method] The heavy manual labeling approach is not scaleable nor novel.

**Strengths Contributions:**

- [Motivation] The online video understanding task is crutial for several real seneraios like robotics, VR and AR, but related benchmarks are still limited. The reviewer consider the effors on this task to be valuable.
- [Novelty] The ideal of using dynamic question answering, where the answer varies as the video streams, is novel and interesting.
- [Method] The heavy manual labeling approach guarantees the quaility of the benchmark.
- [Experiments] The experiments reveals the gap between offline and online video understanding, and demonstrates the limitations of SOTA models on stream video understanding and long sequences.

---

> ### Author Rebuttal · Authors · 2025-07-31
>
> We sincerely appreciate your valuable insights, especially the reminder regarding the dataset's gating issue.
>
> > **w1**: The heavy manual labeling approach is not scaleable nor novel.
>
> **Ans**:Thank you for pointing out the issue. As you mentioned, our annotation work does involve certain costs. Although we designed automated processes during the template collection and validation stages, the core annotation work was performed manually to ensure dataset quality. This step could indeed benefit from further automation to enable scalability.
>
> > about **Ethical Comments**: The authors claim that part of videos are collected form online sources, but no explicit source is described. Whether the usage of these videos aligns with the user policy of such sources needs further verification.
>
> **Ans**:We thank the reviewer for raising this important ethical concern. We confirm that all videos included in our dataset are sourced from publicly accessible online platforms. We have made efforts to ensure compliance with the respective user policies and copyright regulations. To improve transparency, we will include explicit documentation of the video sources and their usage policies in the final version. Furthermore, we are committed to conducting thorough verification to ensure our dataset fully aligns with ethical and legal standards.
>
> > about **Dataset Code Comments & Additional Feedback**: The dataset is hosted on Huggingface. However, the dataset is gated and accessing this dataset requires sharing the contact information (email and huggingface username) with the repository authors, which may break the single-blind revewing process.
>
> **Ans**: We sincerely appreciate your reminder. Due to not carefully reviewing Huggingface's settings, we had enabled the gated access feature, which has now been disabled. We apologize for any inconvenience this may have caused and hope this addresses your concerns.

---

> > ### Comment · Reviewer_SZ3Q · 2025-08-04
> >
> > I thank the authors for their rebuttal. After reading the rebuttal and comments from peer revirewers, I believe my concern has been addressed. Therefore, I will raise my score.

---

> > > ### Author Response · Authors · 2025-08-07
> > >
> > > Thank you for reading our rebuttal. We’re glad our response addressed your concerns, and we truly appreciate your updated evaluation.

---

### Note · Authors · 2025-08-13

Thank you for your efforts in reviewing this paper. Below is the resolution of main concerns and supplementary plans.
## Resolution of main concerns
- **Ungated dataset**: As suggested, HuggingFace settings affecting single-blind review were revised, resolving the concern and led to a score increase..
- **QA annotation process**: Inquiries about the specific details of the QA annotation process have been mentioned both in the responses and the main text. Regarding the issue of the lack of AI assistance in QA annotation, we have also provided a detailed reply based on the idea that dataset quality takes priority.
- **Dataset design**: For the concerns regarding the time-evolving nature of the setting and long-term temporal dependencies, we have specifically pointed out the relevant designs in the dataset, effectively addressing these concerns.
- **Definitions (e.g., "continuous" concept, task taxonomy)**: Responses provide detailed explanations of definitions (with locations in the main text/appendices), resolving these concerns.
- **Evaluating blind language models**: Responses note relevant papers (with citations) have completed such experiments, whose conclusions resolve this concern.
## Revisions to experimental analyses
- **Further analysis of metrics (q0/q1/q2 accuracy)**: Guided by reviewers, we have conducted further analysis on the accuracy of basic and multi-timestamp questions, and discovered relevant inspiring phenomena about online/offline models. At the same time, experiments have confirmed the phenomenon that QA accuracy decreases as time depth increases. This will be added as a supplement to the dataset quality analysis in the official version, thus well addressing the related concerns.
- **Impact of initial perception**: Combined with above analysis, showed its role in enhancing online video model performance. This supports our hypothesis that "strong offline capabilities underpin online abilities" and will be included in revisions.
- **Fine-grained improvements (reasoning step scoring, error analysis)**: We will conduct fine-grained analysis of reasoning steps per reviewers' suggestions; results will be included in the revision upon completion.

In summary, the reviewers' concerns have been largely addressed, and some reviewers have also raised the score for this work. We sincerely thank all authors and reviewers for their contributions to this paper.

---

### Decision · Program_Chairs · 2025-09-18

**Decision:**

Accept (poster)

**Comment:**

The reviews and discussion converged on the view that RTV-Bench addresses an important and timely problem by introducing a benchmark for continuous perception and reasoning in real-time video. Specifically, interesting contributions such as multi-timestamp QA, hierarchical questions, and multidimensional evaluation will be helpful for the video understanding community. Reviewers initially raised concerns about dataset accessibility, the QA annotation process, definitions of “continuous” and task taxonomy, and the adequacy of baseline comparisons. The rebuttal and subsequent discussion clarified most of these issues: the dataset release was corrected, annotation and design choices were explained, and additional analyses were promised or included. While some reviewers remained cautious about the maturity of the benchmark and the completeness of fine-grained reasoning analyses, the consensus acknowledged the novelty, scale, and potential impact of the work. The final decision was guided by the balance b/w the main positive factors (the benchmark’s clear motivation, careful design, and relevance to advancing real-time multimodal learning), and the limiting factors (lingering questions about long-term adoption and the depth of analysis - e.g., on error types). AC believes the merits of this work outweigh the shortcomings, and the paper will be a good addition to the program.

===== FINAL UPDATE FROM DB Track PCs ====

The final decision for this paper has been taken by the program chairs after consultation with the SACs. All Senior Area Chairs have ranked papers according to the feedback from the AC during the review process. We decided to leave the original meta-review to reflect the opinion of the AC in light of the initial discussions with reviewers and SAC.